



# HETEAC – The Hybrid End-To-End Aerosol Classification model for EarthCARE

Ulla Wandinger[1], Athena Augusta Floutsi[1], Holger Baars[1], Moritz Haarig[1], Albert Ansmann[1],
Anja Hünerbein[1], Nicole Docter[2], David Donovan[3], Gerd-Jan van Zadelhoff[3], Shannon Mason[4], and
Jason Cole[5]

[1]Leibniz Institute for Tropospheric Research (TROPOS), Leipzig, Germany
[2]Free University of Berlin (FUB), Institute for Space Science, Berlin, Germany
[3]Royal Netherlands Meteorological Institute (KNMI), De Bilt, The Netherlands
[4]European Centre for Medium Range Weather Forecasts (ECMWF), Reading, United Kingdom
[5]Environment and Climate Change Canada (ECCC), Toronto, Ontario, Canada

**Correspondence:** Ulla Wandinger (ulla.wandinger@tropos.de)

**Abstract.**

The Hybrid End-To-End Aerosol Classification (HETEAC) model for the Earth Clouds, Aerosols and Radiation Explorer (EarthCARE) mission is introduced. The model serves as the common baseline for development, evaluation, and implementation of EarthCARE algorithms. It guarantees the consistency of different aerosol products from the multi-instrument platform and facilitates the conform specification of broad-band optical properties needed for EarthCARE radiative closure assessments.

While the hybrid approach ensures that the theoretical description of aerosol microphysical properties is consistent with the optical properties of the measured aerosol types, the end-to-end model permits the uniform representation of aerosol types in terms of microphysical, optical, and radiative properties. Four basic aerosol components with prescribed microphysical properties are used to compose various natural and anthropogenic aerosols of the troposphere. The components contain weakly

and strongly absorbing fine-mode as well as spherical and non-spherical coarse-mode particles and thus are representative for pollution, smoke, sea salt, and dust, respectively. Their microphysical properties are selected such that a good coverage of the observational phase space of intensive, i.e., concentration-independent, optical aerosol properties derived from EarthCARE measurements is obtained. Mixing rules to calculate optical and radiative properties of any aerosol blend composed of the four basic components are provided. Applications of HETEAC in the generation of test scenes, the development of retrieval algo-

rithms for stand-alone and synergistic aerosol products from EarthCARE's Atmospheric Lidar (ATLID) and Multi-Spectral Imager (MSI), as well as for radiative closure assessments are discussed. In the end, conclusions for future development work are drawn.

## 1 Introduction

The Earth Clouds, Aerosols and Radiation Explorer (EarthCARE) is a joint mission of the European Space Agency (ESA)

and the Japan Aerospace Exploration Agency (JAXA) carrying four sensors, a cloud-profiling radar (CPR), a high-spectral-resolution cloud/aerosol lidar (ATLID), a cloud/aerosol multi-spectral imager (MSI), and a three-view broad-band radiometer



(BBR) (Illingworth et al., 2015; Wehr et al., 2022). Three instruments (ATLID, MSI, and BBR) provide information on the global aerosol distribution and contribute to the overarching EarthCARE goals of sensor synergy and radiation closure with respect to aerosols. The high-spectral-resolution lidar ATLID measures profiles of particle extinction and backscatter coeffi-

cients, lidar ratio, and particle linear depolarization ratio as well as aerosol optical thickness (AOT) at 355 nm along the track of the satellite (Donovan et al., 2022b). The MSI provides AOT at 670 nm (over land and ocean) and 865 nm (over ocean) across a 150 km wide swath (Docter et al., 2022). From combined ATLID and MSI data, the columnar Ångström exponent for the 355-670-865 nm spectral range can be inferred along track (Haarig et al., 2022b). MSI observations are also used to extend the two-dimensional (2D) cross-sections from lidar and radar into the three-dimensional (3D) domain and thus allow

respective 3D radiation modeling (Qu et al., 2022a; Cole et al., 2022). In this way, fluxes, heating rates, and radiances can be calculated and top-of-atmosphere (TOA) radiances and fluxes compared with those derived from BBR measurements (Barker et al., 2022). The EarthCARE aim is to obtain closure of measured and calculated TOA fluxes for a 100 km$^2$ snapshot view of the atmosphere with an accuracy of 10 Wm$^{-2}$, with the final goal to substantially decrease the uncertainties in our knowledge of global radiative forcing (Illingworth et al., 2015; Wehr et al., 2022).

The closure assessments require a proper aerosol classification based on the observations as well as an underlying aerosol model that connects microphysical, optical, and radiative properties of predefined aerosol types to derive the input parameters for radiative transfer calculations. Information on particle size (in terms of effective radius or asymmetry parameter) and on scattering and absorption properties over the relevant spectral range (in terms of wavelength-dependent complex refractive index or single-scattering albedo) is needed. Furthermore, the extinction profile measured with ATLID at 355 nm must be

converted to the visible wavelength range by applying appropriate Ångström exponents, because typically the extinction at a wavelength of 500 to 550 nm is used as input for radiative transfer models. Aerosol classification from the spaceborne observations is also required for quantification of anthropogenic versus natural aerosol loadings of the atmosphere, investigation of aerosol-cloud interaction, as well as assimilation purposes and validation of atmospheric transport models, which carry components like dust, sea salt, smoke, and pollution. Finally, a well-defined aerosol classification model will enable an easier

connection of EarthCARE to previous and upcoming space lidar missions and, in general, helps embed the mission into our understanding of scattering and absorbing aerosols in the climate system (see Li et al., 2022, for an overview).

To facilitate a common aerosol classification throughout the processing chain, and thus the consistency of all EarthCARE aerosol products including those from the radiative closure assessments, the Hybrid End-to-End Aerosol Classification (HETEAC) model has been developed. The model is based on a combined experimental and theoretical (i.e., hybrid) approach and

allows the end-to-end simulation of aerosol properties, from microphysical to optical and radiative parameters of predefined aerosol types. The HETEAC concept was first introduced by Wandinger et al. (2016) and further developed since then. In this paper, we describe the basic considerations, developments, and current applications of HETEAC. The requirements for the EarthCARE aerosol classification scheme are summarized in Sect. 2. Section 3 discusses the idea of the hybrid end-to-end approach in more detail. Section 4 provides the context and heritage of the applied classification scheme. The experimental basis,

on which the typing scheme is based, is briefly summarized in Sect. 5. Major results of the HETEAC model developments are presented in Sect. 6. Examples of the application of HETEAC in the development and implementation of the EarthCARE



processing chain are shown in Sect. 7. Further discussion on the benefits and limitations of HETEAC as well as plans for further developments are provided in Sect. 8.

## 2 Requirements for an EarthCARE aerosol classification scheme

The starting point for the development and implementation of an aerosol classification scheme for EarthCARE was the need to have a common tool that supports the instrument and data synergy and can be used as a baseline for algorithm development and evaluation across the development activities. In the operational phase of the mission, the approach should ensure the consistency of ATLID, MSI, and BBR Level 2a (L2a) and Level 2b (L2b) aerosol products throughout the processing chain (Eisinger et al., 2022) as well as facilitate the consistent specification of broad-band aerosol optical properties necessary for
radiative closure studies.

For the aerosol classification, a suitable set of basic aerosol types must be defined. This basic set should be complete enough to reasonably encompass the range of types encountered in nature, but it should not be more extensive than necessary to keep the number and kind of types traceable throughout different applications. The classification should allow for the separation of natural and anthropogenic aerosols. The types need to be described consistently in terms of microphysical properties (size,
shape, refractive index), which are used to represent them in scattering models, as well as in terms of optical and radiative properties, which are either observed with the EarthCARE instruments and used for the classification (lidar ratio, particle linear depolarization ratio, Ångström exponent) or applied for radiative transfer calculations and closure studies (single-scattering albedo, asymmetry parameter).

The aerosol classification model must be compatible with the EarthCARE End-to-End Simulator (E3SIM, Donovan et al.,
2022a) as the major simulation and implementation test tool for the EarthCARE algorithms. It should provide input parameters for scene simulations with E3SIM and support ATLID, MSI, and BBR retrievals with required a priori information. Therefore, profile as well as columnar observations have to be considered. The model must be able to reproduce existing findings from ground-based and spaceborne observations and should be, as far as possible, consistent with previous approaches, in particular the one of the Cloud-Aerosol Lidar and Infrared Pathfinder Satellite Observations (CALIPSO, Omar et al., 2005, 2009; Kim
et al., 2018; Tackett et al., 2022), to facilitate long-term aerosol studies and trend analysis.

## 3 Hybrid end-to-end concept

The general concept of a hybrid end-to-end aerosol classification scheme is shown in Fig. 1. The red and blue colors stand for the hybrid approach. The aerosol modeling (red) starts from the theoretical description of microphysical particle properties for predefined aerosol types, from which the optical and radiative properties are calculated by applying appropriate scattering
models. The measured optical data (from ATLID and MSI) are the starting point for the experimental part (blue). They are used for aerosol typing, from which microphysical and radiative particle properties follow via parametrization. The circle visualizes the end-to-end concept. The loop is closed by approaching the radiative properties from both sides. The connection




to radiation measurements (from BBR) becomes possible by applying the retrieved parameters in radiative transfer calculations and comparing the modeled and measured values.

The concept shown in Fig. 1 works only, if the red and blue parts of the loop are interlinked. Aerosol typing and parametrization for the retrieval of microphysical and radiative properties from observed optical data are solely based on the underlying model. Therefore, the model must be designed such that the theoretical description of particle microphysics is consistent with experimentally derived optical properties and the observation space is well covered. In this way, a self-contained classification scheme is realized with the proposed hybrid, i.e., combined theoretical and experimental, approach.

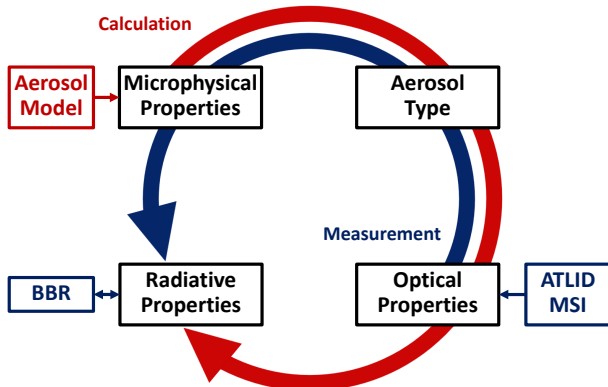

**Figure 1.** Hybrid end-to-end concept for aerosol classification.

While the hybrid end-to-end concept is a general approach, which can be used in various applications, HETEAC is specifically designed for the EarthCARE instrumentation. Table 1 lists the physical quantities used in the four aerosol property groups (see boxes on the circle in Fig. 1). In addition, their role in the EarthCARE retrieval chain is shown. The microphysical model considers particle size, shape, and spectral complex refractive index to allow the modeling of the optical and radiative properties from the ultraviolet (UV) to the far infrared (IR) spectral range as well as the proper coverage of the observation space.

The principles of aerosol typing via the definition of specific aerosol components and their mixtures are discussed in detail in the next sections. The optical properties measured by ATLID and MSI serve as input for the aerosol typing. The radiative properties and the Ångström exponent for the wavelength conversion from 355 to 500 or 550 nm follow from the parametrization according to the model.

## 4   Heritage of aerosol classification used for HETEAC

Prerequisite of any aerosol classification is the selection and definition of aerosol types that are to be identified from the measurements. Aerosol classification has a long history and different approaches are available from the literature. HETEAC developments make use of the heritage of previous attempts, but at the same time consider the specific needs of the EarthCARE mission.





**Table 1.** Aerosol property groups, relevant quantities covered in HETEAC, and their role in the EarthCARE retrieval chain.

| Aerosol property group | Relevant quantities | Role in EarthCARE retrieval chain |
|---|---|---|
| Microphysical properties | Size distribution | Input for typing scheme |
| | Spectral complex refractive index | Input for scattering models |
| | Shape distribution | Input for radiation models |
| | (all per aerosol component) | Input for E3SIM |
| Aerosol type | Fraction of aerosol components | EarthCARE product |
| | (for pure and mixed types) | Output of typing scheme |
| | | Input for MSI retrievals |
| Optical properties | Lidar ratio at 355 nm | EarthCARE products |
| | Particle linear depolarization ratio at 355 nm | Input for typing scheme |
| | Ångström exponent (355-670-865 nm, columnar) | |
| Radiative properties | Effective particle size or asymmetry parameter | Output of typing scheme |
| | Spectral single-scattering albedo | Input for radiation models |
| | Ångström exponent (355-500-550 nm) | Input for closure studies |

One early effort to develop an aerosol classification model is the Optical Properties of Aerosols and Clouds (OPAC) database
(Hess et al., 1998), which in turn builds on the earlier works of Shettle and Fenn (1979), Hänel and Zankl (1979), Deepak and
Gerber (1983), D'Almeida et al. (1991), and Köpke et al. (1997). OPAC allows the construction of aerosol types from a
number of basic aerosol components with well-defined microphysical properties under consideration of hygroscopic particle
growth. Moreover, OPAC provides a comprehensive collection of refractive indexes of basic aerosol components over a wide
wavelength range (0.25–40 $\mu$m). The new version OPAC 4.0 (Koepke et al., 2015) includes a non-spherical description of
dust particles and thus overcomes the previous shortcoming that the optical properties were solely based on Mie scattering
calculations, i.e., only spherical particles could be treated. OPAC has been used in the development of HETEAC as source of
refractive-index information and to study the influence of hygroscopic growth on particle optical properties.

A number of aerosol classification schemes rely on experimental findings from passive and active remote sensing and use
specific optical fingerprints primarily based on intensive, i.e., concentration-independent, optical properties but also on aerosol
load, geographic location, or altitude of occurrence. Such schemes have been developed in the context of the CALIPSO mission
(Omar et al., 2005, 2009; Kim et al., 2018; Tackett et al., 2022), derived from dedicated lidar field studies (e.g., Burton et al.,
2012; Groß et al., 2015) and network measurements (e.g., Nishizawa et al., 2017; Nicolae et al., 2018; Papagiannopoulos et al.,
2018; Floutsi et al., 2022) or retrieved from passive remote-sensing observations (e.g., Russell et al., 2014; Hamill et al., 2016).
Usually, these classification schemes distinguish three groups of aerosol types: 1) so-called pure types like marine, smoke,
pollution, or dust aerosols, 2) mixtures of the former types, and 3) aerosol types that occur only in specific locations or under
specific conditions like the polar regions or the stratosphere. A common understanding exists in defining the pure types of



desert dust, marine aerosol, and anthropogenic pollution. These types can usually be well discriminated because of their well-defined and clearly different optical appearance. Biomass-burning aerosol or smoke is an important player as well, but it has a variable nature and is therefore treated differently. The optical and microphysical properties of smoke vary depending on the kind of burnt material (e.g., Savannah or boreal fires, crop burning), the kind of burning (smoldering or flaming), and the time and height of transport, i.e., the kind of atmospheric processing of particles. The classification schemes also use different ways to account for aerosol mixtures, e.g., mixtures of dust with biomass-burning, marine, or pollution aerosols or mixtures of marine and pollution aerosols.

Nowadays, advanced ground-based instrumentation provides a multitude of parameters from spectral and polarization-sensitive observations with active and passive sensors, which allows for a comprehensive aerosol classification under consideration of multiple aerosol types (e.g., Hamill et al., 2016; Nicolae et al., 2018; Floutsi et al., 2022). In contrast, spaceborne applications are still limited with respect to information content of the measurements. Therefore, more robust approaches based on less sophisticated but reasonable typing schemes are required. For instance, the aerosol project within the ESA Climate Change Initiative (Aerosol_cci) developed a model for passive satellite remote sensing based on four basic aerosol components (Holzer-Popp et al., 2013; de Leeuw et al., 2015). These components comprise two fine particle modes – one strongly absorbing representing smoke and other soot-containing aerosols and one weakly absorbing describing typical anthropogenic emissions – and two coarse particles modes – one with spherical particles characteristic for marine aerosol and one with non-spherical particles such as desert dust. Typical particle sizes and refractive indexes for the four modes were obtained from long-term ground-based Sun photometer observations in the framework of the Aerosol Robotic Network (AERONET, Holben et al., 1998).

For active remote sensing from space, standards have been set by the CALIPSO mission (Omar et al., 2005, 2009). Its version 4 aerosol classification scheme (Kim et al., 2018) considers seven aerosol (sub-)types for the troposphere (marine, clean continental, polluted continental/smoke, desert dust, polluted dust, dusty marine, and elevated smoke). For the stratosphere, the recent update to version 4.5 (Tackett et al., 2022) includes polar stratospheric aerosol, volcanic ash, sulfate, smoke, and unclassified aerosol. CALIPSO aerosol typing relies on lidar Level 1 data, since the methodology has been developed primarily to select proper lidar ratios for Level 2 data retrievals. Thus, selection criteria comprise integrated attenuated backscatter, estimated particle depolarization ratio, and vertical location of the layer as well as the kind of surface above which the observation is made.

The EarthCARE aerosol classification scheme shall preserve the aerosol types of CALIPSO as far as possible to allow long-term global investigations over the lifetime of both missions. However, a more robust typing based on Level 2 data is applied. Thanks to the high-spectral-resolution lidar approach, ATLID retrievals do not require an a priori estimate of the particle lidar ratio, but provide this quantity together with the particle linear depolarization ratio as an observable. EarthCARE's aerosol-type product can thus rely on measured intensive, i.e., concentration-independent, particle properties. For the theoretical description of the microphysical particle properties as part of the hybrid end-to-end concept, the Aerosol_cci approach of using four basic aerosol components has been adopted and modified according to the requirements discussed in Sect. 2. Since ATLID observations are performed at 355 nm and shall be harmonized with the CALIPSO observations at 532 and 1064 nm, an





experimental basis for aerosol typing at the ATLID wavelength and the conversion of results from the UV to the visible (VIS) and near-IR spectral range has been established (Floutsi et al., 2022). The experimental basis and its use in the development of HETEAC is briefly summarized in the next section.

## 5 Experimental basis for aerosol typing

Figure 2 shows a collection of ground-based tropospheric observations of lidar ratio $S$ and particle linear depolarization ratio $\delta$ at 355 nm as well as extinction-related Ångström exponent $\mathring{a}_{\text{ext,UV-VIS}}$ for the 355-to-532 nm wavelength pair. The measurements were taken at various locations in the northern and southern hemisphere between 2006 and 2021. They include contributions from the European Aerosol Research Lidar Network (EARLINET; Pappalardo et al., 2014), PollyNET – the network of lidar systems of type Polly$^{\text{XT}}$ operated by the Leibniz Institute for Tropospheric Research (TROPOS), including mobile systems operated on research vessels (Engelmann et al., 2016; Baars et al., 2016) –, and various field campaigns, in which TROPOS participated over the past 16 years. An early version of the data collection, which served as the starting point for the HETEAC development, was already presented in Illingworth et al. (2015). A detailed description of the extended experimental basis is provided by Floutsi et al. (2022).

The symbols in Fig. 2 are color-coded to distinguish major aerosol types and their mixtures. Orange symbols show observations of dust in different regions of the world, from the Caribbean to Central Asia (e.g., Tesche et al., 2009a; Groß et al., 2011; Haarig et al., 2017a; Hofer et al., 2020). Blue dots indicate measurements in clean marine environments, obtained at Cabo Verde and during cruises with German research vessels across the Atlantic (Groß et al., 2011; Rittmeister et al., 2017; Bohlmann et al., 2018). Black dots represent observations of fresh biomass-burning smoke, most of them taken close to fire spots in the Amazon Rain Forest (Baars et al., 2012) and in South Africa (Giannakaki et al., 2015). Aged biomass-burning smoke, measured after long-range intercontinental transport, is indicated by green squares with black edges (e.g., Haarig et al., 2018). Red symbols stand for pollution (dots) and continental background aerosols (open circles) (e.g., Giannakaki et al., 2015). Different mixtures of these major types such as dust mixed with marine aerosol (e.g., Rittmeister et al., 2017; Bohlmann et al., 2018), pollution, or smoke (e.g., Tesche et al., 2009b; Groß et al., 2011; Kanitz et al., 2014; Giannakaki et al., 2015) are also represented. A complete list of campaigns and references is given in Table 2 of Floutsi et al. (2022). It should be noted that the number of data points in the two panels of Fig. 2 is different, because extinction data at 532 nm to calculate the Ångström exponent were not always available.

From the experimental basis, it can be seen that the discrimination power for the major aerosol types is high in the $S$–$\delta$ space (see Fig. 2a), i.e., the intensive optical properties available from ATLID are well suited for aerosol classification. While the particle linear depolarization ratio allows the identification of dust and dust-containing aerosol mixtures, the lidar ratio is especially useful to distinguish between small absorbing and large non-absorbing spherical particles. Knowledge of the Ångström exponent (see Fig. 2b), which will be available for the atmospheric column by combining ATLID and MSI data, can be helpful for aerosol typing as well but only in combination with additional parameters, since otherwise the ambiguities are very high.





**Figure 2.** Experimental values of (a) lidar ratio and (b) extinction-related Ångström exponent for the 355-to-532 nm wavelength pair versus particle linear depolarization ratio at 355 nm for the troposphere. The symbols indicate individual layer-mean values from selected world-wide measurements with multiwavelength Raman polarization lidars in EARLINET, PollyNET, and during various field campaigns as indicated in the legend to the right of panel (a). The stars show the respective values of the four aerosol components defined in HETEAC, see legend to the right of panel (b).

Fig. 2 also shows the theoretical values for the basic aerosol components defined in HETEAC. They are marked with stars of different color, which relates them to the major aerosol types found from the observations. In the next section, the definition of these basic aerosol components is explained.



# 6 HETEAC model

## 6.1 Definition of basic aerosol components

HETEAC uses four predefined aerosol components to simulate tropospheric aerosol. Similar to the approach of Aerosol_cci, they comprise two fine and two coarse particle modes. The two fine modes consist of either weakly or strongly absorbing spherical particles. One coarse mode contains non-absorbing spherical particles, while the second one is made up of non-spherical particles with wavelength-dependent absorption. The four modes can be interpreted, in an idealized manner, to represent pollution (fine, spherical, weakly absorbing), fresh smoke (fine, spherical, strongly absorbing), marine particles (coarse, spherical, 205 non-absorbing), and dust (coarse, non-spherical, absorbing). More realistic aerosol types can be modeled by mixing these four components. The microphysical properties of the four components and the resulting optical parameters of interest are summarized in Table 2. In the following, a detailed description of the modeling and the selection of the physical parameters for each mode is provided.

**Table 2.** Properties of the four predefined aerosol components in HETEAC ($r_{\text{eff}}$ – effective radius, $r_{0,\text{N}}$ – mode radius of the number size distribution, $r_{0,\text{V}}$ – mode radius of the volume size distribution, $\ln \sigma^*$ – logarithmic mode width, $m_{\text{R}}$ – real part of the refractive index, $m_{\text{I}}$ – imaginary part of the refractive index, $S$ – lidar ratio, $\delta$ – particle linear depolarization ratio, $\mathring{a}_{\text{ext,UV-VIS}}$ – extinction-related 355-to-532 nm Ångström exponent).

| Property | Fine mode weakly absorbing | Fine mode strongly absorbing | Coarse mode spherical | Coarse mode non-spherical |
|---|---|---|---|---|
| $r_{\text{eff}}$, $\mu$m | 0.14 | 0.14 | 1.94 | 1.94 |
| $r_{0,\text{N}}$, $\mu$m | 0.07 | 0.07 | 0.788 | 0.788 |
| $r_{0,\text{V}}$, $\mu$m | 0.1626 | 0.1626 | 2.32 | 2.32 |
| $\ln \sigma^*$ | 0.53 | 0.53 | 0.6 | 0.6 |
| $m_{\text{R}}$ (355 nm) | 1.45 | 1.50 | 1.37 | 1.54 |
| $m_{\text{I}}$ (355 nm) | 0.001 | 0.043 | 4e$-$8 | 0.006 |
| Shape | spherical | spherical | spherical | spheroid |
| $S$ (355 nm), sr | 60.9 | 117.3 | 17.4 | 57.9 |
| $\delta$ (355 nm), % | 0.0 | 0.0 | 0.0 | 25.1 |
| $\mathring{a}_{\text{ext,UV-VIS}}$ | 1.60 | 1.25 | $-0.14$ | $-0.11$ |

## 6.2 Modeling of microphysical properties

A consistent end-to-end aerosol modeling requires the consideration of

– particle size,

– particle shape, and





– complex refractive index.

In HETEAC, each aerosol component is defined by a mono-modal log-normal size distribution of either spherical or spheroid
particles and a wavelength-dependent complex refractive index, which is constant for all particle sizes within the mode.

### 6.2.1  Particle size distribution

The log-normal particle size distribution can be described mathematically in different ways. We provide a brief summary,
which is helpful to quickly compare the parameters used in different models and tools and to prove their consistency in the
EarthCARE processing chain. Two common equations to describe the size distribution are:

$$n(r) = \frac{\mathrm{d}N(r)}{\mathrm{d}r} = \frac{N}{\sqrt{2\pi}r\sigma} \exp\left(-\frac{[\ln(r/r_{0,\mathrm{N}})]^2}{2\sigma^2}\right), \tag{1}$$

$$n(r) = \frac{\mathrm{d}N(r)}{\mathrm{d}r} = \frac{N}{\sqrt{2\pi}r\log\sigma^*\ln 10} \exp\left[-\frac{1}{2}\left(\frac{\log(r/r_{0,\mathrm{N}})}{\log\sigma^*}\right)^2\right]. \tag{2}$$

These equations, which describe the particle number concentration $n$ as a function of particle radius $r$ (with the total particle
number $N$) are completely consistent, but may cause confusion due to the somewhat different description of the mode width
(also called shape parameter of the size distribution). The relation between the logarithmic mode width $\sigma$ (variance) in Eq. (1)
and the mode width $\sigma^*$ (geometric standard deviation) in Eq. (2) is $\sigma = \ln\sigma^*$. It should be noted that some authors use Eq. (1),
but write $\ln\sigma$ instead of $\sigma$. In this case, $\sigma$ indicates the geometric standard deviation and $\ln\sigma$ is the variance. Therefore, the
meaning of $\sigma$ must be carefully checked when different aerosol models are compared.

The mode radius $r_{0,\mathrm{N}}$ is related to the effective radius $r_{\mathrm{eff}}$ of a mono-modal size distribution by

$$r_{\mathrm{eff}} = r_{0,\mathrm{N}}\exp(2.5\sigma^2). \tag{3}$$

Instead of the number size distribution, often the volume size distribution $v(r)$ is applied, i.e., in Eq. (1) and (2) $N$ is replaced
by $V$, and the mode radius $r_{0,\mathrm{V}}$ of the volume size distribution is used, which is calculated from $r_{0,\mathrm{N}}$ as

$$r_{0,\mathrm{V}} = r_{0,\mathrm{N}}10^{3\log^2(\sigma^*)\ln 10} = r_{0,\mathrm{N}}10^{3\sigma^2/\ln 10}. \tag{4}$$

Furthermore, a logarithmic representation is commonly used and thus the size distribution is given as

$$\frac{\mathrm{d}V(r)}{\mathrm{d}\ln r} = \frac{V}{\sqrt{2\pi}\sigma} \exp\left(-\frac{[\ln(r/r_{0,\mathrm{V}})]^2}{2\sigma^2}\right)$$

$$= v(r)\frac{\mathrm{d}N(r)}{\mathrm{d}\ln r} = v(r)r\frac{\mathrm{d}N(r)}{\mathrm{d}r} = v(r)\frac{N}{\sqrt{2\pi}\sigma} \exp\left(-\frac{[\ln(r/r_{0,\mathrm{N}})]^2}{2\sigma^2}\right). \tag{5}$$

E3SIM programming is based on Eq. (1), whereas the OPAC model uses Eq. (2). AERONET retrievals as well as the spheroid
model of Dubovik et al. (2006), which is used below, provide the size distribution in terms of Eq. (5). E3SIM uses the effective
radius $r_{\mathrm{eff}}$ and the variance $\sigma$ as input parameters to describe a mono-modal size distribution and then internally calculates the
radius $r_{0,\mathrm{N}}$ from the effective radius after Eq. (3).





### 6.2.2 Particle shape distribution

For the calculation of polarization-dependent scattering properties, a non-spherical particle shape model together with the respective scattering code is needed. Light scattering by non-spherical particles is a complex topic covered by a wide field of research. In recent years, many efforts have been made to realistically model the shapes of atmospheric particles and calculate their scattering properties (e.g., Gasteiger et al., 2011; Kemppinen et al., 2015; Bi et al., 2018; Saito and Yang, 2021). However, these models are still limited, in particular regarding the maximum particle size, and usually require high computational efforts, which makes them difficult to apply for our purpose. Therefore, as an initial approach for the HETEAC parametrization, we use the traditional way of approximating non-spherical particles by spheroids, i.e., prolates and oblates with a predefined distribution of axis ratios. The axis ratio is defined as the ratio of the length of the axis of rotational symmetry to the length of the axis perpendicular to it, i.e., the axis ratio of prolates is larger than 1 and the one of oblates is less than 1.

Dubovik et al. (2006) provided a pragmatic solution in the form of a spheroid model based on look-up tables (LUTs) of precalculated size- and shape-dependent optical properties of randomly oriented particles. The LUTs cover 25 bins of axis ratios between 0.33 and 3, 41 bins of size parameters between 0.012 and 625 (on a logarithmic-equidistant scale), 22 bins of the real part (1.29–1.7) and 16 bins of the imaginary part ($1 \times 10^{-10}$–0.5) of the refractive index, and 181 scattering angles from 0° to 180°. The model allows the simulation of properties of shape mixtures. The large size range is realized by combining the advanced T-matrix code (Mishchenko and Travis, 1998; Mishchenko et al., 2002) for size parameters up to about 30 with the approximate geometric-optics–integral-equation method (Yang and Liou, 1996) for larger sizes. Dubovik et al. (2006) propose one specific mixture of spheroids that well reproduces the scattering properties of dust measured in the laboratory. Customized mixtures or individual shapes can be used in the model as well. In addition, it is possible to mix spherical and non-spherical particles.

The spheroid model is applied to calculate optical properties of aerosol types containing non-spherical particles in HETEAC. Because the model is very fast, it can be used to study a large variety of parameter combinations and to select appropriate ones.

Regarding the shape distribution, there are two proposals available from the literature, one by Dubovik et al. (2006), which is implemented in various retrieval approaches (e.g., in Aerosol_cci), and one by Koepke et al. (2015), which is used in OPAC 4.0. Whereas the first one is a more or less arbitrary assumption, the latter one follows experimental observations by Kandler et al. (2009) and has already been used to produce the scattering libraries for dust in earlier versions of the EarthCARE End-to-End Simulator (with the old name ECSIM). Figure 3 shows the axis-ratio distributions of the two models. Dubovik's distribution is given for logarithmic-equidistant intervals as used in the spheroid scattering code. The shape distribution of oblates and prolates is symmetric on the logarithmic scale. The distribution used in OPAC 4.0 is provided on a linear scale, but has been interpolated to the logarithmic scale here to make it useable for calculations with Dubovik's code. A discussion of results obtained for these shape distributions is provided in Sect. 6.3.4.

In the context of the shape discussion, it should be noted that the definitions regarding the particle size distribution in Sect. 6.2.1 are based on the particle radius and thus strictly hold for spherical particles only. The size of non-spherical particles is usually described via an equivalent radius (or diameter). Depending on application, equivalence with respect to a sphere of





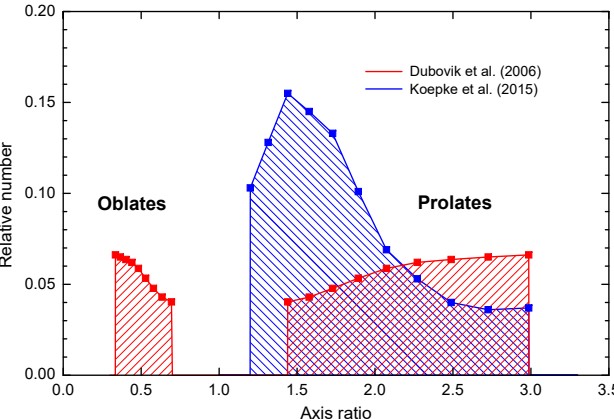

**Figure 3.** Axis ratio distribution proposed by Dubovik et al. (2006), shown in red, and Koepke et al. (2015), shown in blue. The latter one is interpolated from the original linear to the logarithmic grid needed for calculations with Dubovik's code for spheroid scatterers. The logarithmic grid is indicated by the symbols.

the same volume, surface area, or geometrical cross section is used, i.e., a volume-equivalent ($r_{ve}$), surface-area-equivalent ($r_{se}$), or cross-section-equivalent radius ($r_{ce}$) is defined. For randomly oriented convex bodies, the ratio of surface area to average cross section is constant and equal to four (Cauchy's theorem on convex bodies). Thus, $r_{se}/r_{ce} = 1$ as long as we restrict our calculations to spheroids. The ratio of surface area to volume increases with increasing aspect ratio (ratio of major to minor axis) of the spheroid, i.e., $r_{se}/r_{ve} > 1$. Thus, the definition of particle size via parameters like effective radius or size parameter

becomes ambiguous and relations between, e.g., surface-area and volume size distributions developed for spheres do not hold anymore. However, in the case of spheroids with axis ratios between 0.33 and 3, as used here, $r_{se}/r_{ve} < 1.1$. Therefore, in a first approximation, shape effects in the definition of size distribution parameters may be neglected. Nevertheless, one should keep in mind that, e.g., the true effective radius of an ensemble of non-spherical particles is larger (typically 5–10 % for spheroids) than the one given for an ensemble of volume-equivalent spheres.

### 6.2.3 Spectral complex refractive index

The real and imaginary parts of the complex refractive index describe the particles' ability to scatter and absorb electromagnetic radiation, respectively. Strictly speaking, the refractive index is a wavelength-dependent property of a certain material. Particles may be composed of different materials (e.g., dust particles are made up of different minerals), and an ensemble of particles in an atmospheric aerosol probe typically consists of particles with different composition. For the purpose of aerosol modeling,

usually a common (average) refractive index is assumed for all particles of an ensemble. In our case, a wavelength-dependent complex refractive index is assigned to each of the four basic aerosol components.

Figure 4 shows the spectral complex refractive in the wavelength range from 300 to 2250 nm as used for different aerosol components in the OPAC and Aerosol_cci models (Hess et al., 1998; Holzer-Popp et al., 2013). In addition, various measure-

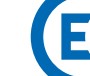


ments of dust refractive indexes in the UV to near-IR range are shown (Kandler et al., 2009; Müller et al., 2009; Petzold et al.,
2009; Di Biagio et al., 2019). The values recommended for the four HETEAC components at the ATLID and MSI measurement wavelengths and at 550 nm are indicated with stars. They are also listed in Table 3. Further discussion and analysis of the selected values is provided for each basic component in Sect. 6.3.

No investigations for wavelengths larger than 2250 nm have been performed in the context of HETEAC developments so far. As mentioned above, the OPAC database provides refractive-index values up to 40 $\mu$m wavelength, which can be used
for broad-band radiative transfer calculations. However, updates similar to the ones performed for the short wavelengths (see discussion below) should be envisaged in view of recent findings, e.g., for dust, as recommended by Di Biagio et al. (2017).

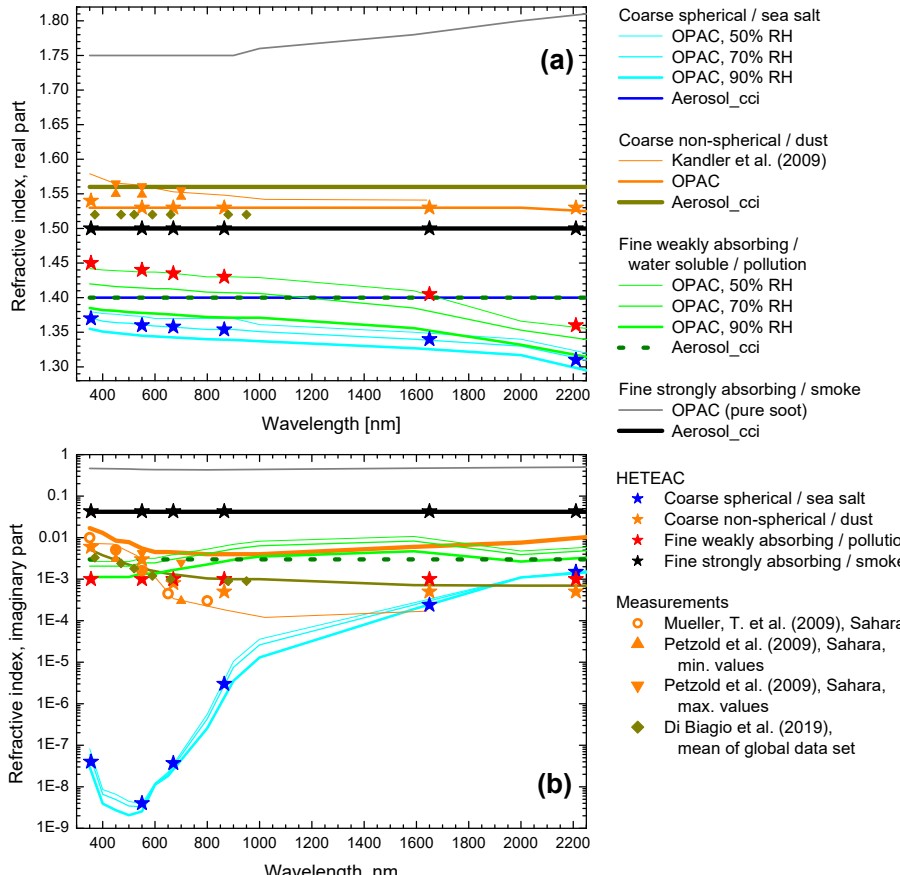

**Figure 4.** Real (a) and imaginary part (b) of the spectral refractive index from different models and measurements. The stars indicate the values selected for HETEAC.





**Table 3.** Complex refractive index of the four predefined aerosol components at selected wavelengths.

| Wavelength | Fine mode weakly absorbing | Fine mode strongly absorbing | Coarse mode spherical | Coarse mode non-spherical |
|---|---|---|---|---|
| 355 nm | $1.450 - 0.001i$ | $1.50 - 0.043i$ | $1.370 - 4.0 \times 10^{-8}i$ | $1.54 - 6.0 \times 10^{-3}i$ |
| 550 nm | $1.440 - 0.001i$ | $1.50 - 0.043i$ | $1.360 - 4.0 \times 10^{-9}i$ | $1.53 - 3.0 \times 10^{-3}i$ |
| 670 nm | $1.435 - 0.001i$ | $1.50 - 0.043i$ | $1.358 - 4.0 \times 10^{-8}i$ | $1.53 - 8.0 \times 10^{-4}i$ |
| 865 nm | $1.430 - 0.001i$ | $1.50 - 0.043i$ | $1.354 - 3.0 \times 10^{-6}i$ | $1.53 - 5.0 \times 10^{-4}i$ |
| 1650 nm | $1.405 - 0.001i$ | $1.50 - 0.043i$ | $1.340 - 2.4 \times 10^{-4}i$ | $1.53 - 5.0 \times 10^{-4}i$ |
| 2210 nm | $1.360 - 0.001i$ | $1.50 - 0.043i$ | $1.310 - 1.5 \times 10^{-3}i$ | $1.53 - 5.0 \times 10^{-4}i$ |

## 6.3 Selection of microphysical parameters for the basic aerosol components

The microphysical properties of the four basic aerosol components have been selected starting from the available knowledge in the literature, as already indicated in Sect. 6.2. The fine-tuning of the parameters was done by comparing the resulting
optical parameters with the experimental basis presented in Sect. 5. As mentioned in Sect. 6.1, the four basic components are considered to mainly represent anthropogenic pollution (small, weakly absorbing particles), fresh biomass-burning aerosol (small, strongly absorbing particles), marine aerosol (large spherical, non-absorbing particles), and mineral dust (large non-spherical, moderately absorbing particles). In the following, the criteria applied in the selection of the microphysical parameters and the consequences regarding the representation of real-world aerosols are discussed.

### 6.3.1 Weakly absorbing fine-mode particles

Fine particulate matter is either directly emitted or generated from precursors by gas-to-particle conversion. Major anthropogenic sources are the combustion of fossil and bio fuels for industrial, transportation, and heating purposes as well as agricultural activities. Anthropogenic aerosol is often modeled as a mixture of water-soluble, i.e., hygroscopic, and insoluble material. The absorption properties can be determined, e.g., via the fraction of insoluble soot contained in the mixture. Com-
pared to smoke from biomass burning (see next paragraph), anthropogenic aerosol is assumed to be weakly to moderately absorbing. Typing schemes use names like continental pollution, industrial pollution, or urban aerosol for the classification of anthropogenic aerosol and sometimes introduce sub-types to distinguish emissions from different sources or regions with different optical properties (e.g., Russell et al., 2014).

For weakly absorbing fine-mode particles in HETEAC, the size distribution from the Aerosol_cci model is adopted (see
Table 2), but the refractive index is modified. Aerosol_cci uses a constant value of $m = 1.40 - 0.003i$ (see Fig. 4, dashed olive lines). In HETEAC, the real part has a slight spectral slope following the OPAC simulations for water-soluble particles at 50 % relative humidity (see red stars and thin green line in Fig. 4a). A constant imaginary part of $m_I = 0.001$ is chosen, i.e., the absorption is reduced compared to the respective component of the Aerosol_cci model. In this way, a better coverage of the





observation space is realized. For instance, the lidar ratio at 355 nm is 78 sr when using the Aerosol_cci refractive index, which
obviously is too high to properly describe the optical properties of polluted continental aerosol (see Fig. 2). The modified
value in HETEAC leads to a more realistic value of 61 sr. If needed, the fine-mode absorption can be increased by mixing of
the weakly absorbing with the strongly absorbing component, which has the same size distribution (see next paragraph and
Sect. 6.4).

Next to absorption, also the size of the particles can change, in particular in dependence on relative humidity. Because
accurate enough humidity information to describe hygroscopic particle growth is not available for spaceborne retrievals, the
effect is not explicitly considered in HETEAC. To study its potential impact on the retrievals, the change of optical properties
in dependence on hygroscopic growth has been investigated with OPAC.

Table 4 compares the optical properties obtained with HETEAC and the Aerosol_cci model for weakly absorbing fine-
mode particles with two representations of continental pollution provided by OPAC. In OPAC, water-soluble (*waso*), insoluble
(*inso*), and soot components (*soot*) are mixed to represented various continental aerosol conditions. Whereas the water-soluble
and soot modes have small mode radii ($r_{0,\mathrm{N}} < 0.05$ $\mu$m, $r_{\mathrm{eff}} < 0.27$ $\mu$m), the insoluble particle mode contains large particles
($r_{0,\mathrm{N}} = 0.47$ $\mu$m, $r_{\mathrm{eff}} = 3.9$ $\mu$m) to account also for (a small fraction of) soil dust and organic or biogenic material in the
continental aerosol. The latter fraction has been omitted in the second case of OPAC simulations (urban) shown in Table 4.
It can be seen that the water-soluble component leads to an increase of the effective radius and to slightly varying optical
properties in dependence on relative humidity due to hygroscopic growth. However, the lidar ratio always remains in a range
of about 60–70 sr, i.e., its sensitivity is low and the value of 61 sr resulting from the microphysical parameters chosen for
HETEAC is a good representation for anthropogenic aerosol independent of actual relative humidity. In addition, it should
be noted that Zieger et al. (2013) found that OPAC tends to overestimate the humidity-growth effects for humidity values of
50–80 %, i.e., the change in optical data for moderate relative humidity may be even smaller than shown in Table 4.

Ångström exponents obtained with HETEAC and the Aerosol_cci model are higher than those calculated with OPAC (see
Table 4). The UV-VIS wavelength pairs used in the simulations are 355 and 532 nm for HETEAC and Aerosol_cci and 350
and 500 nm for OPAC. For VIS-IR, the pairs are 532 and 865 nm and 500 and 800 nm, respectively. Values of about 1.6 in the
UV-VIS and 2.2 in the VIS-IR wavelength range are found for the weakly absorbing fine-mode particles in HETEAC and the
Aerosol_cci model, in good agreement with experimental results. OPAC gives Ångström exponents between 1.0 and 1.2 in the
UV-VIS and between 1.3 and 1.5 in the VIS-IR range for a relative humidity between 50 and 95 %, which are obviously too
low to well represent polluted conditions. The relatively low values for moderate humidity do also not increase significantly,
when the coarse insoluble particles are completely dropped in the simulations (see Table 4). The reason for the low Ångström
exponents lies in the wider size distribution of the fine-mode particles used in OPAC ($\sigma^* = 2.24$ instead of 1.82 for HETEAC
and Aerosol_cci). Thus, for the same effective radius, more optically active large particles on the right wing of the size spectrum
contribute to the scattering properties, without a compensation from the very small and optically inefficient particles on the left
wing. It can be concluded that the size-distribution parameters chosen for the fine mode in HETEAC, by following Aerosol_cci
and thus climatological values from AERONET, provide a better representation of natural conditions than OPAC.





**Table 4.** Comparison of HETEAC, Aerosol_cci, and OPAC model values for effective radius ($r_{eff}$), lidar ratio at 355 nm (350 nm for OPAC, $S_{UV}$), and extinction-related Ångström exponents in the UV-VIS ($\mathring{a}_{ext,UV-VIS}$) and VIS-IR range ($\mathring{a}_{ext,VIS-IR}$) for anthropogenically polluted aerosol.

| Relative humidity | $r_{eff}$, $\mu$m | $S_{UV}$, sr | $\mathring{a}_{ext,UV-VIS}$ | $\mathring{a}_{ext,VIS-IR}$ |
|---|---|---|---|---|
| **HETEAC, fine mode, less absorbing** | | | | |
| | 0.14 | 60.9 | 1.60 | 2.21 |
| **Aerosol_cci, fine mode, less absorbing** | | | | |
| | 0.14 | 78.3 | 1.61 | 2.17 |
| **OPAC, polluted continental, three modes** (*waso, inso, soot*) | | | | |
| 50 % | 0.143 | 59.7 | 1.18 | 1.48 |
| 70 % | 0.150 | 64.3 | 1.16 | 1.47 |
| 80 % | 0.158 | 66.7 | 1.13 | 1.45 |
| 90 % | 0.175 | 69.6 | 1.07 | 1.40 |
| 95 % | 0.198 | 69.8 | 0.99 | 1.33 |
| 98 % | 0.234 | 68.8 | 0.86 | 1.21 |
| 99 % | 0.263 | 65.9 | 0.77 | 1.12 |
| **OPAC, urban, two modes** (*waso, soot*) | | | | |
| 50 % | 0.086 | 63.6 | 1.23 | 1.55 |
| 70 % | 0.097 | 67.5 | 1.21 | 1.53 |
| 80 % | 0.107 | 69.4 | 1.18 | 1.50 |
| 90 % | 0.129 | 71.6 | 1.11 | 1.44 |
| 95 % | 0.156 | 71.3 | 1.02 | 1.36 |
| 98 % | 0.199 | 69.7 | 0.88 | 1.23 |
| 99 % | 0.230 | 66.7 | 0.79 | 1.14 |

### 6.3.2 Strongly absorbing fine-mode particles

As already discussed in Sect. 4, smoke from biomass burning is of variable nature. Its microphysical and optical properties depend on the generation processes (burnt material and kind of fire) as well as on processes during transport in the atmosphere. Smoke is often detected in pronounced lofted atmospheric layers, which can travel over very long distances and remain in the atmosphere for days to weeks. Such smoke plumes may contain not only burnt material but also other aerosols like soil dust taken up during the fire event. Therefore, the modeling of smoke properties is challenging, and mixtures of different components should be taken into account for a realistic representation. Nevertheless, freshly emitted smoke particles are of sub-micron size and contain a high fraction of soot and other absorbing materials, i.e., an absorbing fine mode is required for the description.





HETEAC follows the Aerosol_cci approach and uses a component of strongly absorbing fine-mode particles with the same size distribution as for anthropogenic pollution and a constant refractive index of $m = 1.50 - 0.043i$. These microphysical properties lead to a lidar ratio of 117 sr at 355 nm and an extinction-related UV-VIS Ångström exponent of 1.25 (see Table 2). As can be seen from Fig. 2, the component does not represent typical smoke conditions, but sets an upper limit for the absorption and the resulting lidar ratio covered by the model. More realistic smoke properties can be simulated by mixing the strongly absorbing component with less absorbing fine and coarse particles. If a fraction of non-spherical coarse particles is added, a certain depolarization can be introduced as smoke typically shows linear depolarization ratios between 1 and 10 % in the troposphere (see Fig. 2). Mixing of components is further discussed in Sect. 6.4.

### 6.3.3 Spherical coarse-mode particles

Marine aerosol is primarily composed of water-soluble, coarse sea-salt particles generated by wind-driven physical processes at the ocean surface. Fine-mode particles consisting of non-sea-salt sulfates produced from organic precursor gases contribute to this aerosol type as well. Their number concentrations may be high, but their mass or volume fraction can usually be neglected against the sea-salt component. Except under very dry conditions, marine particles can be assumed to be spherical. The water content and thus size and refractive index of the particles depend on relative humidity. However, also in this case, HETEAC does not consider hygroscopic growth effects explicitly and defines only a typical coarse mode consisting of spherical particles. Again, the size-distribution parameters are taken from the Aerosol_cci model, but the refractive index is modified. While the Aerosol_cci model prescribes a constant value of $m = 1.40 - 0.0i$, HETEAC applies the spectral complex refractive index as suggested by OPAC for a moderate relative humidity of 70 % (see Fig. 4 and Table 3).

As for the weakly absorbing fine-mode particles, a sensitivity analysis regarding the influence of hygroscopic growth on optical parameters was performed with OPAC, and the optical data obtained from the different models were compared. Results are shown in Table 5. In OPAC, clean marine aerosol is defined as a three-modal composition of a water-soluble fine mode (*waso*), a sea-salt accumulation mode (*ssac*), and a sea-salt coarse mode (*sscm*). Refractive index and mode radius of all three modes change with relative humidity. To remove the influence of the fine-mode particles, calculation were also made for the two sea-salt modes only (*ssam, sscm*). From Table 5, it can be seen that the optical parameters vary only slightly in dependence on relative humidity and are all in good agreement with the observations (see Fig. 2). The lidar ratio at 355 nm calculated with OPAC is between 17 and 27 sr, i.e., in the expected range for large spherical, non-absorbing particles. Whereas the Aerosol_cci model gives a relatively small value of 13 sr due to the chosen refractive index, the adoption of the OPAC refractive index in HETEAC leads to a more realistic value of 18 sr.

Extinction-related Ångström exponents from OPAC are slightly negative and similar to the HETEAC and Aerosol_cci values, when only large sea-salt particles are considered. They become slightly positive when a water-soluble fine mode is added to the marine aerosol. Accordingly, also the lidar ratio is somewhat larger in the latter case. Also here, it should be mentioned that Zieger et al. (2013) showed that OPAC may overestimate the humidity-growth effects for moderate values of relative humidity. Nevertheless, it can be seen from Table 5 that the variability of the optical parameters of marine aerosol is anyhow small and on the order of the expected measurement errors. Thus, it is concluded that humidity effects can be neglected in the classification



of marine aerosol from spaceborne lidar observations and that the coarse sea-salt component proposed for HETEAC is a good representation of marine particles.

**Table 5.** Comparison of HETEAC, Aerosol_cci, and OPAC model values for effective radius ($r_{eff}$), lidar ratio at 355 nm (350 nm for OPAC, $S_{UV}$), and extinction-related Ångström exponents in the UV-VIS ($å_{ext,UV-VIS}$) and VIS-IR range ($å_{ext,VIS-IR}$) for marine aerosol.

| Relative humidity | $r_{eff}$, $\mu$m | $S_{UV}$, sr | $å_{ext,UV-VIS}$ | $å_{ext,VIS-IR}$ |
|---|---|---|---|---|
| **HETEAC, coarse mode, spherical** | | | | |
| | 1.94 | 18.1 | $-0.13$ | $-0.20$ |
| **Aerosol_cci, coarse mode, spherical** | | | | |
| | 1.94 | 13.3 | $-0.12$ | $-0.20$ |
| **OPAC, clean marine, three modes (*waso, ssam, sscm*)** | | | | |
| 50 % | 0.803 | 22.7 | 0.14 | 0.13 |
| 70 % | 0.923 | 22.7 | 0.13 | 0.10 |
| 80 % | 1.031 | 26.1 | 0.12 | 0.08 |
| 90 % | 1.267 | 21.0 | 0.12 | 0.06 |
| 95 % | 1.590 | 27.4 | 0.12 | 0.06 |
| 98 % | 2.207 | 23.1 | 0.10 | 0.07 |
| 99 % | 2.836 | 23.2 | 0.08 | 0.06 |
| **OPAC, sea salt, two modes (*ssam, sscm*)** | | | | |
| 50 % | 1.227 | 18.7 | $-0.16$ | $-0.09$ |
| 70 % | 1.379 | 18.6 | $-0.16$ | $-0.12$ |
| 80 % | 1.516 | 21.7 | $-0.16$ | $-0.13$ |
| 90 % | 1.808 | 17.0 | $-0.14$ | $-0.15$ |
| 95 % | 2.197 | 22.8 | $-0.13$ | $-0.16$ |
| 98 % | 2.902 | 19.1 | $-0.10$ | $-0.12$ |
| 99 % | 3.575 | 19.8 | $-0.08$ | $-0.10$ |

### 6.3.4   Non-spherical coarse-mode particles

Mineral dust is the aerosol with the highest abundance in the atmosphere. Most of the dust is emitted from deserts along the
northern hemispheric dust belt, reaching from northern Africa with the Sahara as the largest global dust source over Middle East, Central Asia to China and Mongolia. Also deserts in North and South America, southern Africa and Australia contribute to the global dust load. Dust particles are of non-spherical shape, have a relatively large size, and their mineral composition varies depending on source region. Since the optical properties sensitively depend on actual size, shape, and complex refractive index, a proper selection of dust microphysical parameters for HETEAC is challenging, not only because of the natural variability of
dust properties but also because of the limitations in the modeling of non-spherical particles (see Sect. 6.2.2). Therefore, a large





number of simulations have been performed under consideration of a wide range of assumptions and parameters proposed in the literature.

Some representative results of these studies are shown in Fig. 5. The simulations have been performed with Dubovik's code for the two spheroid distributions shown in Fig. 3. Panels (a) and (b) of Fig. 5 show results at 355 nm for the size distribution
proposed by Aerosol_cci, i.e., a constant effective radius of 1.94 $\mu$m, and a range of complex refractive indexes (orange symbols). As indicated in Fig. 4, mineral dust exhibits a considerably higher absorption in the UV than in the VIS and IR spectral range. The refractive index depends on mineral composition, and a variety of values can be found in the literature. A value of $1.56 - 0.005i$ at 355 nm has been proposed by Aerosol_cci, which leads to the $S$–$\delta$ positions encircled in red. OPAC (Koepke et al., 2015) proposes $1.53 - 0.017i$, i.e., a much higher imaginary part and thus a very strong absorption, resulting in
unrealistically high lidar ratios of 150–200 sr for the two distributions (green circles). Kandler et al. (2009) calculated a value of $1.58 - 0.007i$ at 355 nm by analyzing the mass contributions of different minerals in dust samples from the Western Sahara (blue circles). Di Biagio et al. (2019) derived complex refractive indexes from 19 samples collected in eight source regions worldwide. Their values range from 1.49 to 1.54 in real part (over the entire 370–950 $\mu$m wavelength range) and from 0.0011 to 0.0088 in imaginary part at 370 nm, with a global mean value of $1.52 - 0.0033i$ in the UV (cyan circles). By comparing
panels (a) and (b) of Fig. 5 with the experimental basis in Fig. 2, it can be seen that with the spheroid distribution proposed in OPAC 4.0, which follows from the measurements provided by Kandler et al. (2009), a good agreement between measured and simulated values for a realistic range of refractive indexes is found, while the spheroid distribution after Dubovik et al. (2006) leads to much higher lidar ratios than typically observed.

In panel (c) of Fig. 5, a subset of values from panel (b) ($r_{\mathrm{eff}} = 1.94$ $\mu$m, $m_{\mathrm{R}} = 1.50$–1.58, $m_{\mathrm{I}} = 0.002$–0.007) is shown
and compared with results for varying effective radius at three selected values of refractive index. The dark brown symbols span a relatively wide range of effective radii from 0.5 to 3.1 $\mu$m. AERONET data indicate that the effective radius of the dust coarse mode is typically between 1 and 2.7 $\mu$m (see, e.g., Holzer-Popp et al., 2013, Fig. 1 and 2). In this size range, for constant refractive index, an increase of the effective radius leads to an increase of the lidar ratio and a decrease of the linear depolarization ratio. In general, we see that for the assumed spheroid distribution, a variation of real and imaginary
part of refractive index and of effective radius shifts the $S$–$\delta$ points in the diagram in three different directions and, thus, the natural variability around typical values of $S = 50$ sr and $\delta = 25\,\%$ as found from observations can be well covered with the simulations.

Even if the model seems to work fine at the wavelength of 355 nm, we have to be careful regarding the spectral behavior of the simulated values. Figure 6 shows Ångström exponents calculated for the wavelength pair 355 and 532 nm. As in
Fig. 5b, the spheroid distribution after Koepke et al. (2015) and an effective radius of 1.94 $\mu$m was used. The refractive index at 355 nm was fixed to $m = 1.54 - 0.006i$. The extinction-related and backscatter-related Ångström exponents were then computed for a varying refractive index at 532 nm. In all cases, negative values of the Ångström exponent are found. While the extinction-related Ångström exponent is insensitive to changes of the refractive index and almost constant at $-0.1$, the backscatter-related Ångström exponent is strongly dependent on the refractive index and shows large negative values.
For instance, for the refractive index of $1.53 - 0.003i$ chosen for HETEAC (see Fig. 4 and Table 3), the backscatter-related



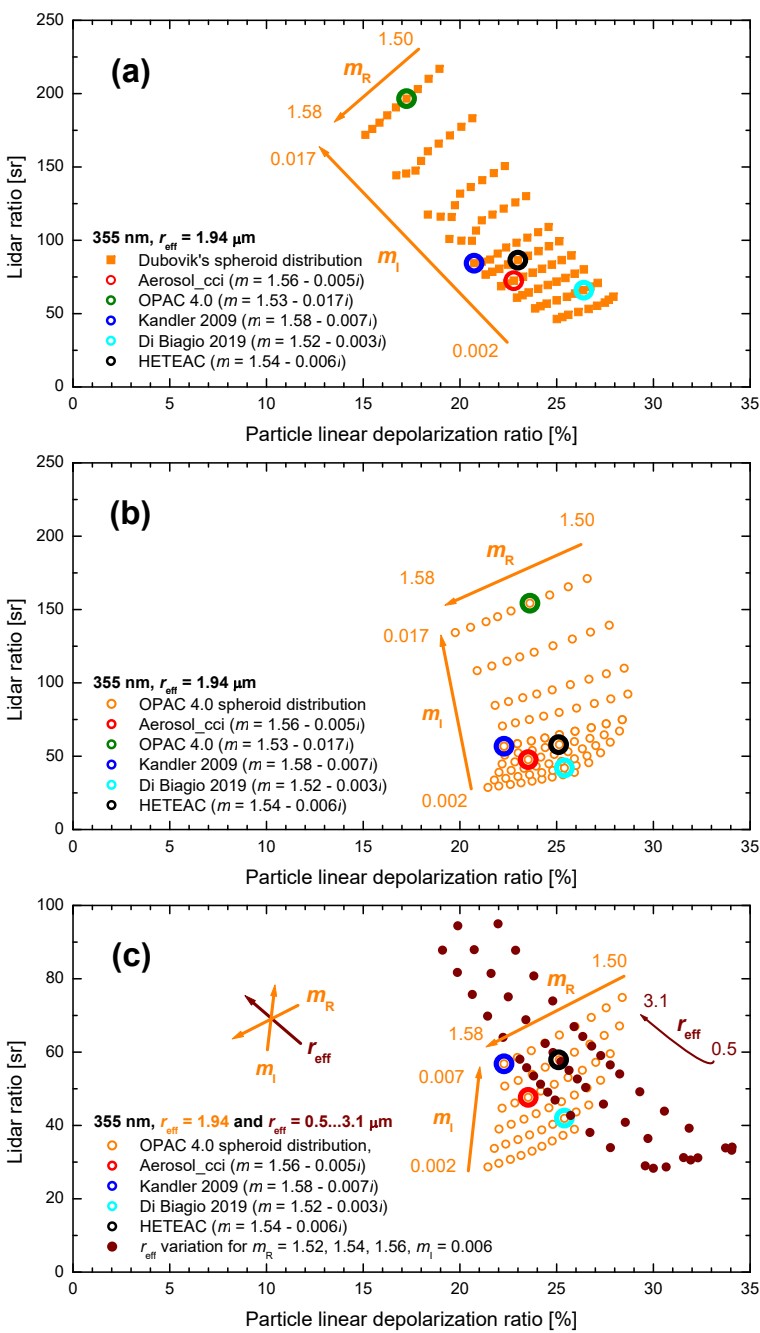

**Figure 5.** Simulated values of lidar ratio versus particle linear depolarization ratio at 355 nm for (a) the spheroid distribution after Dubovik et al. (2006) and (b and c) the spheroid distribution after Koepke et al. (2015). Orange symbols show results for a constant effective radius of 1.94 $\mu$m and varying complex refractive index as indicated in the figure. Dark brown symbols in panel (c) show results for varying effective radius.



Ångström exponent becomes $-1.6$. As a consequence, the modeled lidar ratio at 532 nm is much smaller than at 355 nm. For the parameters selected for HETEAC, we obtain 58 sr at 355 nm and 31 sr at 532 m. In principle, negative dust Ångström exponents are not unusual in nature, but values below $-1$ seem unrealistic. For instance, Veselovskii et al. (2020) reported backscatter-related Ångström exponents down to $-0.75$ for measurements in West Africa and also discussed the dependence

of the values on the wavelength-dependent imaginary part of the refractive index. The observational mean values from our experimental basis are typically close to 0, with slightly positive extinction-related Ångström exponents of 0.1 and 0.2 and slightly negative backscatter-related Ångström exponents of 0.0 and $-0.2$ for Saharan and Central Asian dust, respectively (Floutsi et al., 2022). Accordingly, the measured lidar ratios at the two wavelengths are similar, with values of about 53 sr for Saharan dust at both wavelengths and 43 and 38 sr for Central Asian dust at 355 and 532 nm, respectively (Floutsi et al.,

2022). The model can reproduce this spectral behavior only, when unrealistically low real parts and high imaginary parts of refractive index at 532 nm are assumed (see values in the upper left corner of Fig. 6). We argue that the spheroidal particle shape model is not able to fully mimic the scattering properties of irregularly shaped dust particles at 180°. Specific features for exact backscattering, similar as known for spheres, could play a role. These limitations may be overcome in future with more realistic scattering models for irregularly shaped particles.

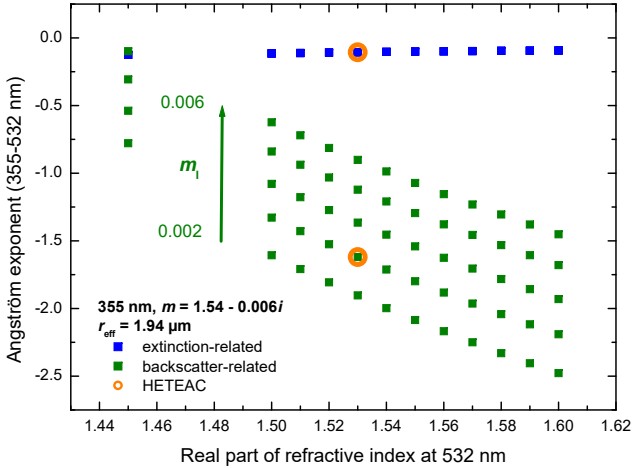

**Figure 6.** Simulated values of the extinction-related (blue) and backscatter-related Ångström exponents (green) for the 355-to-532 nm wavelength pair. The spheroid distribution after Koepke et al. (2015), the effective radius of 1.94 $\mu$m, and the refractive index of $m = 1.54 - 0.006i$ at 355 nm are kept constant, while the refractive index at 532 nm is varied as indicated.

Keeping the limitations of the scattering model for non-spherical particles in mind, we can conclude that a satisfying representation of dust is obtained in HETEAC with the parameter settings for non-spherical coarse particles as provided in Table 2 and under consideration of the experimentally derived spheroid distribution as proposed for OPAC 4.0 (Koepke et al., 2015). Humidity-growth effects do not play a role for dust particles and can be neglected (Denjean et al., 2015). The selected refractive index of $m = 1.54 - 0.006i$ is tuned to best represent observations of Saharan dust as the most abundant type (see Fig. 2).





For specific studies, such as radiative closure assessments, it might be necessary to adapt the refractive index according to the source region.

## 6.4    Definition of component mixtures

As shown above, major aerosol types like anthropogenic pollution, fresh smoke, sea salt, and dust can be described with four basic aerosol components consisting of two fine and two coarse particle modes, each with predefined particle shape
(distribution) and complex refractive index. The four components define the optical parameter space (corner points) that is covered with the model. The microphysical properties of the components have been selected such that an optimum overlap between model and observation space in terms of optical data is obtained. By mixing the four basic components, we can fill the model space in between the corner points and thus compose new or mixed aerosol types in accordance with the observations.

   For this purpose, we assume an external mixture of the particles and apply mixing rules to obtain the intensive optical
parameters of multimodal aerosol compositions. Starting from the microphysical parameters of each component, the individual scattering properties per unit particle volume (e.g., 1 $\mu\mathrm{m}^3\,\mathrm{cm}^{-3}$) are calculated with the respective scattering model first. Then, the optical parameters of interest are derived from the extinction, scattering, and backscatter coefficients per unit volume ($\overline{\alpha}_i$, $\overline{\beta}_i$, and $\overline{\sigma}_i$, respectively), the depolarization ratio $\delta_i$, and the relative volume contribution $v_i$ of each mode $i$. In this way, we obtain, e.g., the lidar ratio

$$S = \frac{\sum_i v_i \overline{\alpha}_i}{\sum_i v_i \overline{\beta}_i} \tag{6}$$

and the particle linear depolarization ratio

$$\delta = \frac{\sum_i v_i \overline{\beta}_i \frac{\delta_i}{1+\delta_i}}{\sum_i v_i \overline{\beta}_i \frac{1}{1+\delta_i}} \tag{7}$$

of the mixture.

   Figure 7 shows the $S$–$\delta$ diagram at 355 nm for bimodal and trimodal mixtures. The stars indicate the pure components.
The numbers stand for the volume mixing ratio in percent. It can be seen that the depolarization ratio of dust (orange stars) sensitively reacts to the addition of non-depolarizing particles. The dependence is non-linear in terms of particle volume contribution, i.e., the mixing of dust with fine-mode particles or sea salt can be well resolved as long as the dust contribution dominates. Vice versa, a relatively large amount of dust is needed to cause a considerable depolarization ratio. Similarly, very large and very small lidar ratios are only obtained when the absorbing or the sea-salt components dominate, respectively. Many
of the mixtures produce lidar ratios of 50–70 sr and particle linear depolarization ratios below 5 %. Such values are indeed most often observed in nature and are typical for polluted continental sites (see Fig. 2).



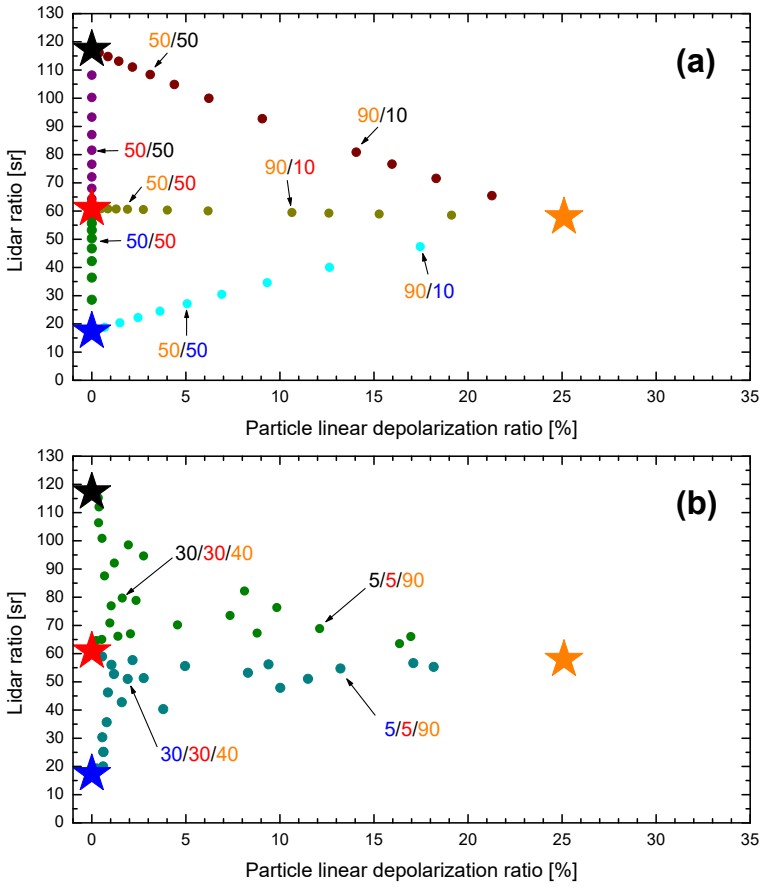

**Figure 7.** Simulated values of lidar ratio versus particle linear depolarization ratio at 355 nm for (a) mixtures of two components and (b) mixtures of three components. The pure components are indicated by stars (red = fine mode, weakly absorbing; black = fine mode, strongly absorbing; blue = coarse mode, spherical; orange = coarse mode, non-spherical), and their properties are given in Table 2. The numbers in the plots show the mixing state in terms of particle volume in percent (colors as for the stars).

## 6.5 Modeling of radiative properties

If the aerosol typing is used for radiation studies, the radiative properties of the aerosol components and their mixtures are needed. Radiative transfer models typically require input information on particle size (e.g., in terms of effective radius or

asymmetry parameter) and on scattering and absorption properties (i.e., complex refractive index or single-scattering albedo) over the relevant spectral range. Furthermore, in the case of EarthCARE, the extinction profile measured with ATLID at 355 nm must be converted to the input wavelength used in the model (e.g., 550 nm) by applying appropriate (vertically resolved) Ångström exponents.




To support radiative transfer calculations, HETEAC provides LUTs of radiative parameters for the four pure aerosol components and their mixtures. Again, external mixing rules are applied to calculate the Ångström exponent

$$\mathring{a} = \frac{\ln\left(\sum_i v_i \overline{\alpha}_{i,\lambda_1} / \sum_i v_i \overline{\alpha}_{i,\lambda_2}\right)}{\ln\left(\lambda_2/\lambda_1\right)},\tag{8}$$

the single-scattering albedo

$$\omega_0 = \frac{\sum_i v_i \overline{\sigma}_i}{\sum_i v_i \overline{\alpha}_i},\tag{9}$$

and the asymmetry parameter

$$g = \frac{\sum_i v_i \overline{\sigma}_i g_i}{\sum_i v_i \overline{\sigma}_i}.\tag{10}$$

The calculations are performed for lidar wavelengths of 355, 532, and 1064 nm and imager wavelengths of 670, 865, 1650, and 2210 nm, as well as for 550 nm. The parameters are given for 314 mixtures with equally distributed volume fractions of the components of 0 %, 5 %, 10 %, 20 %, 30 %, ..., 90 %, 95 %, 100 %. Results for the 355-to-670 nm Ångström exponent as well as for the single-scattering albedo and asymmetry parameter at 550 nm are shown in Fig. 8. The values are color-coded and projected to the $S$–$\delta$ diagram to illustrate how the typing via the lidar measurements can be used to select proper data for radiative transfer calculations.

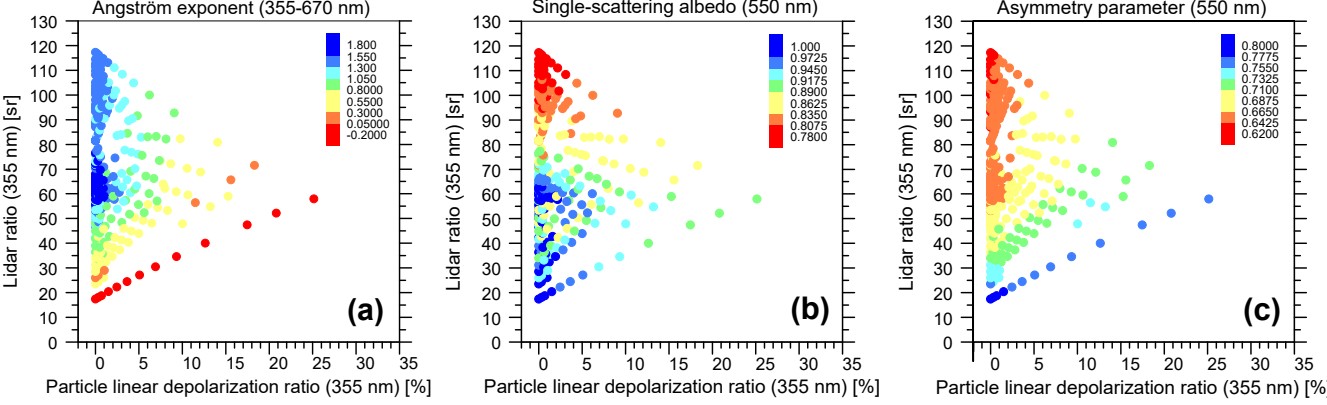

**Figure 8.** Simulated values of (a) Ångström exponent for the 355-to-670 nm spectral range, (b) single-scattering albedo at 550 nm, and (c) asymmetry paramter at 550 nm. The radiative properties are color-coded in the $S$–$\delta$ plane at 355 nm (see legends).





## 7 Applications of HETEAC

HETEAC is or will be applied in the development of EarthCARE retrieval algorithms, the generation of test scenes for algorithm performance evaluation, as well as for data evaluation and radiation closure studies. In the following, a brief overview on applications that are important for the preparation of the mission is given. In Sect. 7.1, the use of HETEAC in the generation of E3SIM aerosol scenes is explained, while Sect. 7.2 shows its application in the retrieval of various EarthCARE stand-alone and synergistic aerosol products.

### 7.1 Simulation of aerosol test scenes

Artificial atmospheric scenes have been extensively used to test the performance of the EarthCARE algorithms during their development. For this purpose, three dedicated scenes (entitled *Halifax*, *Baja*, and *Hawaii*) were generated based on output of the Global Environmental Multiscale (GEM) model (Qu et al., 2022b; Donovan et al., 2022a). Each scene represents a typical EarthCARE frame of about 5000 km length, corresponding to one eighth of an entire orbit, which is the standard for EarthCARE data processing (Eisinger et al., 2022). Since GEM does not provide aerosol forecasts, supplementary information was taken from the Copernicus Atmosphere Monitoring Service (CAMS) model. The CAMS aerosol component fields, containing sea salt, dust, organic and black carbon, and sulfate aerosol, were mapped in an ad-hoc fashion to the HETEAC components of coarse spherical, coarse non-spherical, strongly absorbing fine-mode, and weakly absorbing fine-mode particles, respectively. This procedure was successful in creating aerosol fields in E3SIM with a realistic range of extinction, lidar ratio, and AOT at the ATLID and MSI wavelengths, which were then applied to test the stand-alone and synergistic aerosol retrievals (e.g., Donovan et al., 2022b; Wandinger et al., 2022; Docter et al., 2022; Haarig et al., 2022b; Mason et al., 2022). An example is provided in the next section.

### 7.2 Aerosol classification in EarthCARE retrievals

#### 7.2.1 ATLID aerosol retrievals: the A-TC product

The ATLID Target Classification (A-TC) algorithm, which produces the corresponding A-TC product, is part of the ATLID L2a Profiles (A-PRO) processor and is explained in detail by Donovan et al. (2022b). The A-TC approach works with aerosol types (i.e., target classes for aerosol) that are described as mixtures of the four basic HETEAC aerosol components. The joint $S$–$\delta$ distribution for each aerosol type is modeled using a Gaussian probability distribution defined by a mean lidar ratio, particle linear depolarization ratio, their associated Gaussian widths and correlation. Figure 9 shows the $S$–$\delta$ probability distribution functions (PDFs) schematically. Based on the HETEAC information on the $S$–$\delta$ distribution for pure and mixed components (Fig. 7) and in agreement with the experimental basis (Fig. 2), six aerosol types corresponding to marine aerosol, continental pollution, smoke, dust, dusty smoke, and dusty aerosol mixtures are defined such that the PDFs span the same phase space as the HETEAC model. At the same time, they are well separated from the expected distribution for ice crystals, i.e., the methodology also supports aerosol-cloud discrimination. The volume and extinction mixing ratios for the six aerosol types





used in the algorithm developments are presented in Table 6. These values will be further refined based on algorithm tests with experimental data in the future.

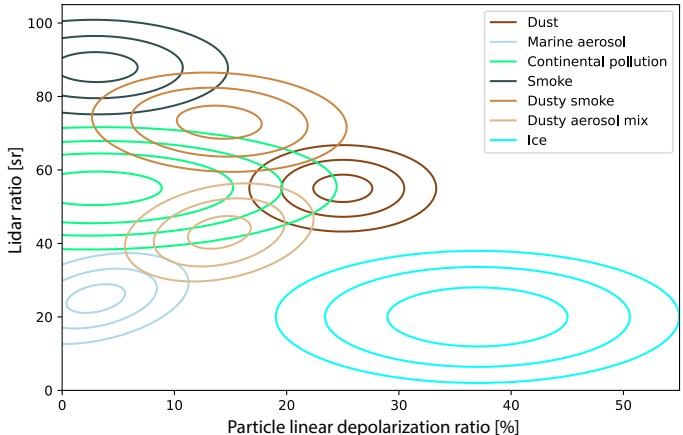

**Figure 9.** Schematic representation of the $S$–$\delta$ probability distribution functions used to determine the aerosol-related elements of the A-TC product.

**Table 6.** Volume mixing ratios (VMR) and extinction mixing ratios (EMR) corresponding to the six tropospheric A-TC aerosol types shown in Fig. 9.

| | Fine mode weakly absorbing | | Fine mode strongly absorbing | | Coarse mode spherical | | Coarse mode non-spherical | |
|---|---|---|---|---|---|---|---|---|
| | VMR | EMR | VMR | EMR | VMR | EMR | VMR | EMR |
| Dust | 0.015 | 0.13 | 0.00 | 0.00 | 0.02 | 0.02 | 0.965 | 0.85 |
| Marine aerosol | 0.00 | 0.00 | 0.00 | 0.00 | 0.99 | 0.99 | 0.01 | 0.01 |
| Continental pollution | 0.35 | 0.85 | 0.00 | 0.00 | 0.54 | 0.12 | 0.10 | 0.02 |
| Smoke | 0.19 | 0.22 | 0.59 | 0.76 | 0.00 | 0.00 | 0.21 | 0.02 |
| Dusty smoke | 0.00 | 0.00 | 0.12 | 0.61 | 0.00 | 0.00 | 0.88 | 0.39 |
| Dusty aerosol mix | 0.05 | 0.36 | 0.00 | 0.00 | 0.40 | 0.26 | 0.55 | 0.38 |

An example aerosol classification result is shown for the *Halifax* scene in Fig. 10. According to the CAMS output, sulfate and sea-salt aerosols were present in the atmosphere. Respective extinction fields of weakly absorbing fine-mode and spherical coarse-mode particles were generated with E3SIM, which represent the model truth for the aerosol typing (lowermost two left panels). The entire extinction field of the *Halifax* scene including clouds, aerosols, and precipitation is shown in the upper left panel. By comparing the A-TC results (bottom-right panel) with the model truth, it can be seen that, overall, the A-TC
results capture the presence of two distinct main aerosol types in the southern segment (right part) of the frame. In the northern segment, the aerosols are largely obscured by the extended areas of high cloud coverage. Misclassifications and unknown-type





determinations are related to noise and errors in the input A-PRO lidar ratios and particle linear depolarization ratios used by the classification procedure (uppermost two right panels). These input data are stored in the ATLID Extinction, Backscatter and Depolarization (A-EBD) product Donovan et al. (2022b).

The aerosol classification from A-TC is reproduced in the ATLID–CPR synergistic target classification product (AC-TC, Irbah et al., 2022) later in the processing chain. There, the only modifications occur in rare instances where CPR detections may help distinguish ice from aerosols. AC-TC is the basis for further synergistic retrievals explained in Sect. 7.2.5.

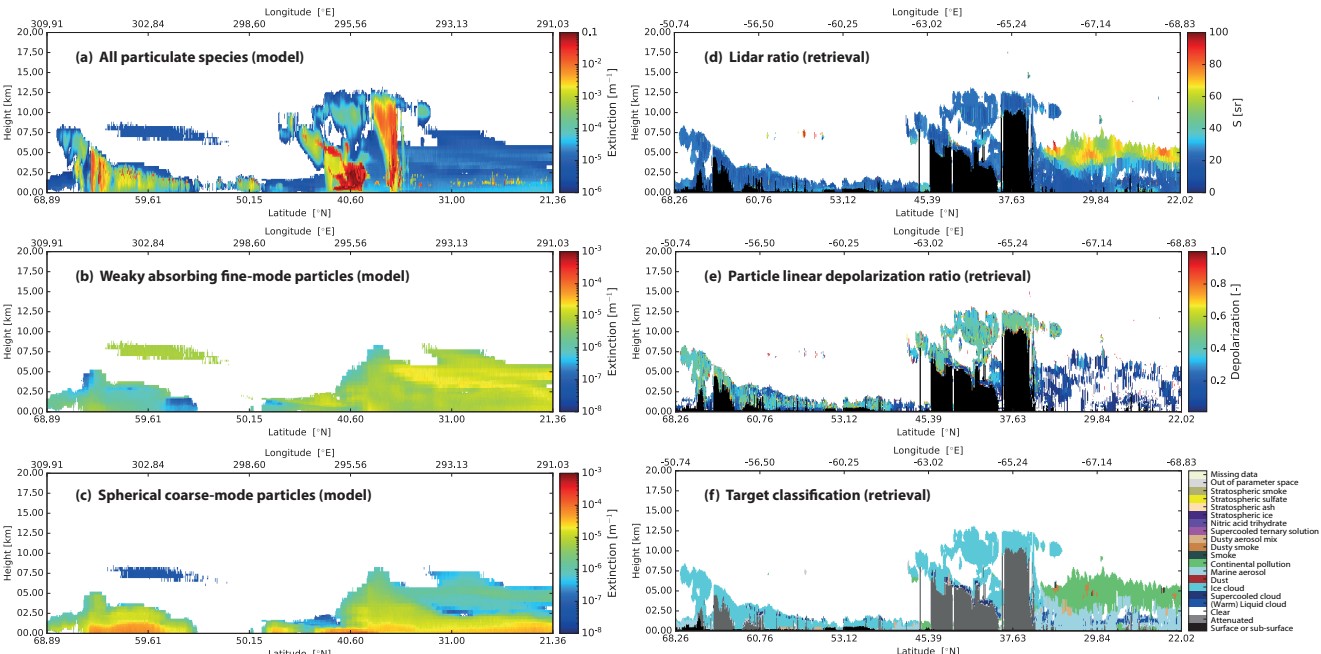

**Figure 10.** Aerosol classification for the *Halifax* scene. Panel (a) shows the modeled nadir total extinction field including all aerosol, cloud, and precipitation types. Panels (b) and (c) display the modeled extinction fields for weakly absorbing fine-mode particles (the primary component of the A-TC type "continental pollution") and spherical coarse-mode particles (the primary component of the A-TC type "marine aerosol"), respectively. Panels (d) and (e) show the fields of lidar ratio and particle linear depolarization ratio, respectively, retrieved with A-PRO and stored in the A-EBD product. These values are used together with the predefined $S-\delta$ PDFs (shown in Fig. 9) to determine the aerosol-related elements of the A-TC product depicted in panel (f). Note that the longitude is defined in the interval [0° E, 360° E] in the model (left panels) and [−180° E, 180° E] in the retrieval (right panels).

### 7.2.2   ATLID aerosol retrievals: the A-ALD product

The ATLID Aerosol Layer Descriptor (A-ALD) algorithm, which generates the A-ALD product, is part of the ATLID L2a
Layer Products (A-LAY) processor described in Wandinger et al. (2022). The algorithm detects aerosol layer boundaries and calculates the layer-mean optical properties such as backscatter coefficient, extinction coefficient, lidar ratio, and particle linear





depolarization ratio. The latter two might be used as input for an aerosol classification per layer based on HETEAC in a similar way as done in the A-TC algorithm (Donovan et al., 2022b). However, in the A-LAY processor a different approach is implemented to prepare the synergy with MSI, for which columnar values are needed (see Sect. 7.2.3). Thus, column-

integrated aerosol classification probabilities are derived from the A-TC product. For this purpose, the relative contribution of each of the six A-TC aerosol types to the total AOT at 355 nm is calculated by weighting the probability of occurrence for each height bin with the respective extinction coefficient at 355 nm and integrating this information over the entire profile. The column-integrated aerosol classification probabilities can then be compared with the aerosol-type results of the MSI retrieval (see Sect. 7.2.3 and 7.2.4).

### 7.2.3 MSI aerosol retrievals: the M-AOT product

The MSI L2a Aerosol Optical Thickness (M-AOT) retrieval, which produces the M-AOT product (Docter et al., 2022), uses an approach based on LUTs for its forward model. The LUTs rely on radiative transfer simulations with the Matrix Operator model MOMO (Hollstein and Fischer, 2012; Fell and Fischer, 2001), for which the optical properties of the four HETEAC components are used. To allow for the presence of more than one pure HETEAC component within a column, the four components have been

additionally mixed via their contribution to aerosol optical thickness. For this purpose, 25 HETEAC-based component mixtures have been defined. The choice of the most appropriate mixture to be used in the retrieval is then based on climatological knowledge over land and on the best-fitting mixing above ocean (Docter et al., 2022).

While ATLID-based retrievals provide direct information on the type or mixtures, the imager-based retrieval is not able to do so due to the limited information available from MSI. At best, the M-AOT retrieval can distinguish between coarse and

fine modes and respective mixtures over ocean, based on the spectral behavior of AOT determined from the four bands of the so-called VNS (visible, near-infrared, and short-wave infrared) camera of MSI and the low surface contribution to the signal in off-glint regions. On the other hand, the classification of the sub-types sea salt (spherical) or dust (non-spherical) and weakly absorbing or strongly absorbing fine mode is much more difficult, because of the similarity in the optical properties that are accessible with MSI measurements. Hence, the retrieved M-AOT aerosol optical thickness is always accompanied by the used

HETEAC aerosol component mixture. In this way, users may be able to apply ad-hoc corrections to M-AOT estimates at 670 and 865 nm.

### 7.2.4 Synergistic aerosol retrievals: the AM-ACD product

The ATLID–MSI Aerosol Column Descriptor (AM-ACD) is produced by the ATLID–MSI L2b Column Products (AM-COL) processor, which is described in detail by Haarig et al. (2022b). The algorithm compares the aerosol typing of ATLID (A-TC,

Sect. 7.2.1; A-ALD, Sect. 7.2.2) and MSI (M-AOT, Sect. 7.2.3) along the satellite track and thus provides a quality check and additional information for users with respect to the limited MSI information. To facilitate the comparison, the dominant aerosol type of each typing scheme is determined. The dominant aerosol type is defined by the highest column-integrated aerosol classification probability (ATLID) and the component with highest contribution to the aerosol mixing ratio (MSI), respectively. The four pure types of A-TC are dominated by the four aerosol components defined in HETEAC and used in M-



AOT, respectively. Thus, their contributions can be directly compared. The challenge arises from the two mixed types in A-TC, "dusty mix" and "dusty smoke", which have to be converted back into the basic HETEAC components. Such a conversion can be done with the help of the mixing rules described in Sect. 6.4 and the PDFs, which define mixed types. Details and further discussion are provided in Haarig et al. (2022b).

### 7.2.5 Synergistic aerosol retrievals: the ACM-CAP product

The ATLID–CPR–MSI L2b Cloud, Aerosol and Precipitation (ACM-CAP) processor, which generates the respective ACM-CAP product, carries out a retrieval of the aerosol classes identified in the synergistic target classification (AC-TC, see last paragraph of Sect. 7.2.1), constrained by the synergy of ATLID and MSI solar and thermal IR channels. The ACM-CAP algorithm takes a different approach from those taken in A-PRO, in that the size distributions and physical properties of the aerosol types, including their lidar ratio, are predetermined entirely by the A-TC classification (i.e., by the volumetric mixtures

of the four pure HETEAC components given in Table 6). Only the profile of particle number concentration is retrieved for each aerosol class, which scales the retrieved extinction and aerosol optical depth. To constrain the retrieval of horizontally homogeneous aerosol fields from inherently noisy lidar measurements at the scale of the joint standard grid, a Kalman smoother is applied such that the retrieved quantities in each profile are constrained by the values in adjacent profiles. A more detailed description and evaluation of ACM-CAP aerosol retrieval is given in Mason et al. (2022).

### 7.2.6 Radiative closure assessments: the ACM-RT product

Inclusion of aerosols into EarthCARE's radiative closure assessments (Barker et al., 2022) requires they be specified in the forward radiative transfer calculations (Cole et al., 2022). The specification is done following the HETEAC model. For the basic aerosol components, specific extinction (for solar), specific absorption (for thermal), single-scattering albedo, and asymmetry parameter are computed for 166 wavelengths between 0.2 and 400 $\mu m$. The single-scattering properties are then combined

into the aerosol types using external mixing and volume mixing ratios according to Table 6. The single-scattering albedo and asymmetry parameter for the mixtures are computed following Eq. (9) and (10), respectively. Rather than use specific extinction and absorption for each wavelength and mixture, values for each mixture are normalized by the mixture extinction at 355 nm. This normalization allows these to be scaled by the ATLID extinction profile (Donovan et al., 2022b) when computing the optical properties for the radiative transfer calculations.

The radiative transfer models used to generate the ATLID–CPR–MSI Radiative Transfer (ACM-RT) product work with optical properties averaged over wavelength intervals (Cole et al., 2022). Averaging of the aerosol single-scattering optics over the intervals was done using wavelength-specific weighting. Weightings for solar wavelength intervals were downwelling irradiances averaged at the tropopause and surface from line-by-line data (Iacono et al., 2008) for a tropical atmosphere at a solar zenith angle of zero degrees. For thermal wavelength intervals, weightings were the Planck function at 275 K.



## 8 Conclusions

We have developed an aerosol classification model for the EarthCARE mission, which serves as the common baseline for development, evaluation, and implementation of algorithms and can be used for the exploitation of measurement data later on. The major feature of the model is the consistent end-to-end description of particle microphysical, optical, and radiative properties. The model supports aerosol typing with ATLID and MSI and can be applied for radiation closure assessments by using BBR measurements but also other spaceborne or surface data. Based on the heritage of previous typing approaches and an advanced experimental data base from ground-based lidar measurements at multiple wavelengths, four basic aerosol components containing weakly and strongly absorbing fine-mode as well as spherical and non-spherical coarse-mode particles were selected to describe the aerosol microphysical properties. These components can be used to compose the major aerosol types of anthropogenic pollution, smoke, marine aerosol, and dust as well as their mixtures. Size, shape, and refractive-index parameters of the components were thoroughly adjusted to assure that the modeled optical properties cover the expected observational phase space, in particular in terms of the EarthCARE observables lidar ratio, particle linear depolarization ratio, and Ångström exponent. Mixing of the components allows the simulation of a wide range of natural conditions. In this way, it is possible to link the optical fingerprints delivered by the spaceborne instruments to certain pure and mixed aerosol types and to assign respective radiative properties to the observed scenes.

Mixing rules in HETEAC are based on the assumption of external aerosol mixtures, i.e., the different particles maintain their individual physical and chemical properties while being located in the same scattering volume. This assumption is well justified in many cases, in particular for the mixing of coarse and fine particles such as dust and smoke or sea salt and pollution, and is reflected in typical bimodal or multi-modal size distributions obtained from in situ and remote-sensing measurements. Nevertheless, effects of internal mixing should be kept in mind. As already discussed in Sect. 6.3.2, particles originating from combustion processes undergo chemical and physical processing during and after emission. They change their properties over their lifetime and may be composed of soluble and insoluble materials. Such an internal structure is often accounted for by applying a core-shell model for the calculation of optical properties, i.e., the particle is modeled as consisting of a spherical, insoluble, absorbing core surrounded by a liquid solution. A major result of such investigations is the enhancement of absorption, and thus the decrease of the single-scattering albedo, when the same amount of black carbon is assumed to be contained as a core within a water-soluble shell instead of making up a separate fraction of particles (e.g., Jacobson, 2000, 2001; Lesins et al., 2002; Cappa et al., 2012; Li et al., 2022). However, Cappa et al. (2012) also showed that this effect may be overestimated by the idealized spherical geometry and is probably less pronounced in reality.

Another aspect that has to be considered regarding the mixing state is its influence on the hygroscopic growth of particles. The studies with OPAC presented in Sect. 6.3.1 and 6.3.3 are based on external mixing of soluble and insoluble components and showed that changes in the lidar ratio due to water uptake may be on the order of 10–20 % and thus play a minor role for aerosol typing based on HETEAC. Veselovskii et al. (2020) reported similar changes of the 355 nm lidar ratio for African biomass-burning aerosol. They observed increasing values from 62–80 sr for increasing relative humidity from 25–85 %, which could be well explained with the modeled behavior of hygroscopic, absorbing, homogeneous spheres. Düsing et al.





(2021) determined higher lidar-ratio enhancement factors by applying a core-shell scattering model to measured in situ data.
For continental European aerosol, they found an increase of the 355 nm lidar ratio by a factor of 1.3 and 1.6, when the relative
humidity increased from 50 to 80 % and from 50 to 90 %, respectively.

The investigations on the effects of internal mixing described above are limited to case studies and the methods have large
uncertainties. Thus, it is difficult to draw general conclusions for HETEAC. Overall, the assumption of an external aerosol
mixture may not always be appropriate. However, the discussed effects of enhanced absorption and hygroscopic growth are
mainly related to internal mixing of fine-mode particles and thus contribute together with all other chemical and physical
variations to their overall bulk appearance in the atmosphere, for which the model is actually designed. EarthCARE data with
their limited information content will only allow the identification of more and less absorbing fine-mode aerosols and the
discrimination of coarse-mode aerosols. For this purpose, the approach of external mixing is appropriate, sufficiently robust,
and well supported by the experimental data from ground-based observations.

Major ambiguities of aerosol typing by using the $S$–$\delta$ phase space at the single ATLID wavelength of 355 nm result from
the similarity of optical fingerprints with low particle linear depolarization ratio ($\delta < 5\,\%$) and median lidar ratio ($S$ between
40 and 70 sr). Such values can be caused by different particle blends and will in particular impede a clear separation of aged
tropospheric smoke and continental pollution. As a consequence, a proper assignment of the single-scattering albedo for radia-
tive transfer calculations and closure assessments is difficult (see Fig. 8b). Additional criteria such as geographical location and
altitude of aerosol layers, supported by source analysis, are required for an advancement of the typing in these cases and may
be considered in future upgrades of the EarthCARE retrieval schemes. In general, an in-depth validation with ground-based and
airborne observations is needed after the launch of the mission to evaluate the aerosol classification results and their applica-
tion in radiative closure studies. Ground-based lidar instruments that measure extinction, backscattering, and depolarization at
multiple wavelengths (typically 355, 532, and 1064 nm) and ideally also have a fluorescence detection capability for the iden-
tification of organic materials (contained in smoke and biogenic particles) are best suited to provide a comprehensive aerosol
classification for validation purposes, because they are able to resolve the ambiguities that result from the limited information
content of the ATLID measurements at a single wavelength.

Further improvements of HETEAC are desirable for the modeling of non-spherical particles. In general, a more realistic
representation of particle shapes in scattering models is an urgent issue to answer open questions on relations between dust
microphysical and optical properties. Experimental studies on relationships between lidar ratio and dust composition (i.e.,
refractive index, see Schuster et al., 2012; Veselovskii et al., 2020) are based on retrievals that make use of the spheroid
scattering model and shape distribution provided by Dubovik et al. (2006). As shown in Sect. 6.3.4, the model does not properly
reflect the backscatter spectral behavior of natural dust. Discrepancies between dust optical properties directly measured with
lidar at three wavelengths and those derived from AERONET observations by applying the spheroid model were also reported
by Haarig et al. (2022a). Thus, care must be taken in interpreting any results that rely on the application of a scattering model in
the retrieval of dust properties. It should always be investigated whether obtained relationships are caused by natural phenomena
or artificially induced by model-inherent dependencies and a priori assumptions. In Sect. 6.3.4, similar as in Veselovskii et al.
(2020), dependencies of the $S$–$\delta$ relationship on the complex refractive index have been investigated under the assumption of



a fixed particle size and shape distribution. In addition, it was shown that changes in particle size and shape distribution have a strong influence on the observed values and typically mask the effects of mineral composition (see Fig. 5). The importance of a realistic representation of particle shape for the modeling of lidar-derived dust optical properties has already been emphasized by Gasteiger et al. (2011). Since then, several model studies underlined the sensitivity of $S$ and $\delta$ on particle size and shape parameters, including surface roughness (e.g., Kemppinen et al., 2015; Bi et al., 2018; Saito and Yang, 2021; Kong et al., 2022). However, a comprehensive picture under consideration of various natural conditions is missing. In the end, improvements of HETEAC for a better representation of dust particles will require further research based on a strong effort of combining field and laboratory studies to evaluate potential scattering and shape models against real-world observations. Size-, shape-, and composition-dependent spectral backscattering measurements at exactly 180° in the laboratory, such as introduced by Miffre et al. (2022), are a prerequisite for the success of this work.

The four basic aerosol components and their mixtures considered in HETEAC do not cover very specific categories of particles that may occur in the atmosphere under certain conditions. Some examples are included in the experimental data base (Floutsi et al., 2022) and show where further ambiguities and misclassifications can happen. For instance, sea-salt particles change their shape from spherical to cubic under dry conditions. Therefore, thin layers of depolarizing particles are sometimes observed at the top of the marine boundary layer, when it is in contact with the dry free troposphere (Haarig et al., 2017b). Since the depolarization ratio of cubic salt particles is much smaller (around 8 % at 355 nm) than that of dust, the model will describe them as a mixture of spherical and non-spherical coarse-mode particles. Because of the ambiguity in the optical parameters, identification of dry sea salt would require an additional screening of atmospheric scenes for these specific layers. Yet, it is unclear whether the resolution of ATLID is good enough to detect them at all. Similarly, freshly emitted volcanic ash particles in the troposphere may be interpreted as dust because of their high depolarization ratio. Even if the few available observations indicate that ash could be separated from dust by a higher depolarization ratio (see Floutsi et al., 2022, Fig. 2), the number of available observations is too sparse and the microphysical representation in the model too uncertain to define an extra ash component in HETEAC at the moment. Other aerosols that may need extra treatment in applications are Arctic haze, which consists of strongly aged particles with a characteristic size distribution, or biogenic particles like pollen, which can also show dust-like fingerprints due to their large size and non-spherical shape. Whether it is worthwhile to include such specific types in the EarthCARE aerosol classification scheme can only be decided by evaluating the quality and information content of the EarthCARE data in the course of the mission.

So far, HETEAC focuses on tropospheric aerosols. Nevertheless, the ATLID L2a processors are able to deal with stratospheric aerosol as well and preliminary typing categories are considered in the A-TC product (see Fig. 10). As for the troposphere, the categorization is based on two-dimensional Gaussian distributions of typical $S$ and $\delta$ values known from the literature. Mean values of 55 sr and 45 % for volcanic ash, 40 sr and 3 % for sulfate aerosol, and 70 sr and 3 % for stratospheric smoke are considered in this typing scheme. As shown, e.g., by Ansmann et al. (2021) and Floutsi et al. (2022), the identification of stratospheric smoke, which has considerably different properties than tropospheric smoke, is challenging. Therefore, the stratospheric aerosol classification for EarthCARE needs further investigations and a full end-to-end implementation in HETEAC. For this purpose, it is planned to develop a HETEAC 2.0 version before the launch of EarthCARE. In general,



HETEAC will be updated regularly based on EarthCARE validation studies, which will be performed during the entire lifetime
of the mission.

*Author contributions.* UW has developed the HETEAC model and drafted the manuscript. AAF, HB, MH, and AA worked on the experimental lidar data base and contributed to the definition of the model aerosol components. AAF created the look-up tables for aerosol mixtures. DD and GvZ constructed the HETEAC-based scattering libraries for E3SIM and implemented the aerosol typing scheme in the A-PRO processor. ND, AH, and UW worked on the MSI-related aerosol typing issues. MH implemented aerosol typing parameters in the A-LAY and AM-COL processors. SM and JC contributed with the parts related to the ACM-CAP and ACM-RT processors, respectively. All authors were involved in the discussions during the HETEAC development and contributed material and/or text to the manuscript.

*Competing interests.* UW is member of the editorial board of Atmospheric Measurement Techniques and co-editor of the Special Issue to which this paper contributes. The peer-review process was guided by an independent editor. The authors have no other competing interests to declare.

*Acknowledgements.* This work has been funded by ESA grants 4000112018/14/NL/CT (APRIL) and 4000134661/21/NL/AD (CARDINAL). Oleg Dubovik kindly provided the spheroid scattering model used in the developments and Stefan Horn helped in setting up the calculations. We thank Tobias Wehr and Michael Eisinger for their continuous support over many years and the EarthCARE developer team for valuable discussions in various meetings.





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
