# Peer review of "HETEAC – The Hybrid End-To-End Aerosol Classification model for EarthCARE"

_EGUsphere, 2022_

## Referee Comment (RC2)

Review of HETEAC – The Hybrid End-To-End Aerosol Classification model for EarthCARE by Wandinger et al.

This is a very nice paper describing the model to be used for tropospheric aerosol classification associated with the ATLID and MSI measurements to be acquired during the EarthCARE mission. The paper provides a good description of how the aerosol components were determined, the aerosol optical and physical characteristics of these components, and how they are combined and used to provide aerosol types from the ATLID measurements. The manuscript provides several Figures and Tables to describe these items and an example applied to simulated data. The model should be a good starting point for the EarthCARE aerosol classification. The authors correctly note that this scheme should preserve the CALIOP aerosol types as much as possible in order to facilitate long-term investigations using data from both missions. Along those lines, as noted in comments below, it would be helpful if additional information were provided to determine how the aerosol optical and microphysical characteristics of aerosol types associated with ATLID classification compare with those associated with the CALIOP aerosol types. I recommend publication of this manuscript after addressing the minor comments listed below.

1. Lines 35-45 and discussion of Figure 8. How much variation or uncertainty in aerosol size, absorption (SSA), asymmetry parameter, Angstrom exponent is associated within the aerosol classification for a particular set of lidar observables (lidar ratio, depolarization)? How does this uncertainty then impact the closure of TOA fluxes to the desired accuracy of 10 W/m$^2$?

2. Lines 44 and 525 mention aerosol components associated with aerosol transport models (e.g. sulfate, organic carbon, black carbon). Line 525 mentions that these aerosol components in the CAMS model were mapped to the HETEAC components. How was this mapping done? It would be helpful to know this mapping in order to use the EarthCARE measurements to help evaluate the ability of aerosol transport models to apportion aerosol extinction and AOD to the model aerosol components.

3. Line 166. Why were only ground-based observations of the lidar ratio and particle linear depolarization used without any airborne measurements?

4. Figure 2. It seems that the HETEAC fine strongly absorbing component provides an upper bound on the observed lidar ratio, the HETEAC coarse spherical component provides a lower bound on the observed lidar ratio, and both components provide lower bounds on the observed particle linear depolarization. However, there seems to be several observations of particle linear depolarization ratio that exceed the depolarization ratio of the HETEAC coarse non-spherical component. Could/should this component have been chosen such that its depolarization ratio was higher (~30%) to better bound the observed depolarization ratios?

5. Table 2. It would be helpful to know the values of lidar ratio, linear particle depolarization ratio, and Angstrom exponent corresponding to the standard Nd:YAG wavelengths (532, 1064 nm) used by many aerosol lidars including CALIOP.

6. Figure 4. Do the HETEAC component values of refractive index shown in Figure 4 correspond to dry (RH=0%?, RH<40%) conditions?

7. Line 330. How accurate would RH have to be in order to be considered in HETEAC? Are there any plans to use model reanalysis RH fields when producing updated versions of the ATLID data products?

8. Table 5. This table doesn't show dependence of linear particle depolarization on RH. Should we assume that depolarization for these particles is negligible for all RH?

9. Line 400 and lines 705-712. The HETEAC components do not account for nonspherical sea salt. The conclusion mentions the presence of these aerosols in thin layers near the top of the marine BL (Haarig et al., 2017b) and gives the impression that these aerosols are not considered in the HETEAC scheme because: a) they are not observed often and only in thin layers near the BL top, b) the depolarization ratio of these particles is relatively small (~8% at 355 nm), and c) ATLID may have insufficient resolution to detect these aerosols. An article was recently submitted to Frontiers in Remote Sensing that will indicate that items a), b), and perhaps c) are not necessarily true so the suggestion is made to begin considering how the HETEAC model would deal with such aerosols in the future.

10. Lines 445-465. From this discussion, it seems that the HETEAC model for coarse, non-spherical aerosols (dust) can correctly reproduce the lidar observations of lidar ratio and linear depolarization ratio at 355 nm but would be unable to do so at 532 nm? Is this correct? If so, how will this impact the use of the HETEAC scheme when relating ATLID observations of dust to those from CALIOP?

11. Figure 7. How does the HETEAC handle lidar observations of lidar ratio and/or particle depolarization that lie outside the triangles of points formed by the HETEAC components?

12. Figure 9 and Table 6 provide lidar observable values that correspond to the six tropospheric A-TC aerosol types and how these types are comprised of the HETEAC components. Lines 154-155 state that "The EarthCARE aerosol classification scheme shall preserve the aerosol types of CALIPSO as far as possible to allow long-term global investigations over the lifetime of both missions." This is a commendable goal and will be very important as one would expect that researchers will attempt to combine the CALIOP and ATLID measurements for long term records. The six tropospheric A-TC aerosol types seem to coincide with some of the same aerosol types (at least in name) associated with the CALIOP tropospheric aerosol types. For these apparently common types, how do the lidar observables (at 532 nm) associated with the A-TC types coincide with the corresponding CALIOP values observed (particle depolarization) or assumed (lidar ratio) for these associated aerosol types? How do the underlying aerosol properties (e.g. size, SSA, refractive index) for this common aerosol types compare between A-TC and CALIOP aerosol types?

13. Figure 10. This is a nice figure to show an example of the HETEAC aerosol classification and brings to mind a few questions.
    a. I don't recall seeing elsewhere in the paper a discussion of the spatial and vertical resolutions of the ATLID retrievals of lidar ratio and linear particle depolarization; what are these resolutions? Presumably the resolutions associated with the lidar ratio retrievals are coarser than the resolutions associated with the retrievals of particle linear depolarization. If this is true, how does the HETEAC classification deal with these different resolutions?
    b. There are gaps (white areas) in the lidar ratio and depolarization images on the right. Are these gaps because the aerosol loading was too small for trustworthy measurements? If so, what are the minimum aerosol extinction values for which there will be expected retrievals of lidar ratio and linear particle depolarization? Likewise,

what are the minimum aerosol extinction values required for the HETEAC aerosol classification?

c. Looking again at the images on the right, there are more white areas (no retrievals?) associated with linear particle depolarization than for lidar ratio. Why? (I would have expected the opposite). Does the HETEAC algorithm still attempt to perform a classification if there is a retrieval of only one of two lidar observables?

d. The right side of the top right image shows relatively high values (70-80 sr) of the lidar ratio and low values of aerosol depolarization around 5 km. Figure 2a shows that these lidar ratios and depolarization ratios are associated are smoke and/or mixtures that contain smoke. However, the target classification shown in the bottom right seems to show this area is dominated by continental pollution with little, if any, smoke or smoke mixtures. Why?

14. Line 565. Can the column integrated aerosol classification probabilities be illustrated for the example shown in Figure 10?

15. Line 678. Suggest changing to "Ground-based and airborne measurements that measure…"

---

## Author Comment (AC1)

**Reply to the comments**

We thank the two anonymous referees for their kind evaluation of the manuscript. The comments helped us add necessary information and clarifications. Please find our answers (blue text) to the comments (black text) below. Respective changes to the manuscript are indicated in blue in the version attached to this reply. Line numbers in our answers refer to this manuscript version. Tracked changes do not include minor corrections of spelling/grammar not related to the referee comments.

**Anonymous Referee #1**

Overall this is a solid paper describing the HETEAC aerosol optics/mixing model that is used to aid in EarthCARE algorithm, development. As the authors note, EarthCARE is unique among satellite systems in its high degree of multi sensor integration in deriving aerosol products. Therefore, algorithm developers need a beginning baseline to ensure all of the systems are using compatible models. In and of itself, I think the paper does exactly what it sets out to do: explaining the model, the rationale for why they made the decisions they did, and provide some theoretical uncertainties. From a science reviewer point of view, there is not much to say. Given this is a baseline, they are keeping it all rather simple at this point. They are projecting against the major sources in a logical manner and compare well with observations and existing optical models such as OPAC.

I only suggest minor revisions in a few areas. Going through the paper, I made many notes when the authors massively oversimplified the discussion. I am ok with a simple model as necessity dictates, but the authors need to be clear what the implications of the simplifications are so users can sequester uncertainty. However, the conclusion covers nearly all of my distress (no longer necessary to write them down here)-especially on mixing state which I was most concerned about (which makes the section not really the conclusions). So the paper is at times at odds between declarative statements made throughout, and then in the conclusion waving the hands. Therefore, my only suggestion is to move a lot of that material forward, and perhaps make it an early section. State essentially that for necessity, you have a simple model, but simple models are, well, simple. That is ok with what you want to do.

We thank the referee for bringing these incongruities up. It was indeed a difficult decision where to put which discussion when writing the paper. Simplifications and their implications need to be explained, of course, but we also didn't like to disturb the flow in describing the model itself too much. As a compromise, we have now introduced an additional discussion section (Sect. 8), separated from the shorter conclusion and outlook section (Sect. 9). In Sect. 8, we indicate the different aspects of the discussion by introducing respective subsections, and we also added additional points that were brought up by the referees. We refer to the new Sect. 8 at corresponding places in the paper to make the reader aware of the further discussion.

Other minor things that the authors may want to emphasize is

1) these models probably will not be great for haze events such as in China or India;

This is true. We included the aspect in the discussion in Sect. 8.2 (lines 683-684).

2) use of the lidar ratio is dependent on having a good retrieval, which for lower concentrations is not a given;

The reviewer is correct here, the lidar ratio can only be used for typing when it is retrieved with useful accuracy. For aerosols, roughly speaking, it is expected that useful layer-average lidar-ratio retrievals (SNR of the retrieval better than 100 %) can be done for minimum extinction values on the order of 5.0e-6 m$^{-1}$ on the 50-100 km horizontal scale. A detailed description of the applied averaging schemes is provided in the paper by Donovan et al.: "The ATLID L2a profile processor (A-AER, A-EBD, A-TC and A-ICE products)", which will be submitted to the Special Issue soon. We added some text in Sect. 7.2.1 (lines 567-572, see also reply to comment 13 of Referee #2).

3) practically "dusty smoke" have long been the CALIOP's go to for "We don't know what this is really." How will you handle such uncertainty across platforms.

As described in more detailed below (see reply to comment 12 of Referee #2) , we believe that it will take quite some effort to harmonize the long-term data sets. Indeed, the "dusty smoke" class is similar to the "polluted dust" type defined for CALIPSO, and it mainly stands for mixtures of dust with aerosols from combustion processes. Similarly, the "dusty mix" in A-TC and the "dusty marine" type of CALIPSO represent a mixture of dust with weakly or non-absorbing coarse-mode particles, in particular sea salt. The challenge for the harmonization will be to set the boundaries between the pure and the mixed types appropriately to allow for a comparible classification. Here, we will need support from long-term ground-based validation measurements at multiple wavelengths (to cover both ATLID and CALIOP) as well as statistical evaluations of the long-term data sets to adjust them to each other. We added some discussion on the required future efforts in the conclusion and outlook section (lines 757-771).

Other than these I wish the authors well.

Thank you very much!

**Anonymous Referee #2**

This is a very nice paper describing the model to be used for tropospheric aerosol classification associated with the ATLID and MSI measurements to be acquired during the EarthCARE mission. The paper provides a good description of how the aerosol components were determined, the aerosol optical and physical characteristics of these components, and how they are combined and used to provide aerosol types from the ATLID measurements. The manuscript provides several Figures and Tables to describe these items and an example applied to simulated data. The model should be a good starting point for the EarthCARE aerosol classification. The authors correctly note that this scheme should preserve the CALIOP aerosol types as much as possible in order to facilitate long-term investigations using data from both missions. Along those lines, as noted in comments below, it would be helpful if additional information were provided to determine how the aerosol optical and microphysical characteristics of aerosol types associated with ATLID classification compare with those associated with the CALIOP aerosol types. I recommend publication of this manuscript after addressing the minor comments listed below.

We thank the referee for this positive evaluation. We try to address all comments below. We would like to emphasize that some of the comments, although being of high relevance for the mission, go beyond the basic Level 2 algorithm development work supported by ESA and presented in this Special Issue. We are very much looking forward to working with the community on the validation and application of EarthCARE algorithms and products once the mission is in space, as well as on respective improvements of the methodologies later on.

1. Lines 35-45 and discussion of Figure 8. How much variation or uncertainty in aerosol size, absorption (SSA), asymmetry parameter, Angstrom exponent is associated within the aerosol classification for a particular set of lidar observables (lidar ratio, depolarization)? How does this uncertainty then impact the closure of TOA fluxes to the desired accuracy of 10 W/m2?

These questions go to the heart of the EarthCARE mission and belong to the radiation closure assessments to be done as a major science activity of EarthCARE. There is no simple answer to the questions. Variations/uncertainties in microphysical and optical aerosol properties do not translate one to one into changes of radiative properties and TOA fluxes. As can be seen from Fig. 8, in some cases the influence of variations will be low, while in other cases we have to deal with ambiguities and may need auxiliary information to reduce the errors in the radiative properties. Other points to be considered are the layering and/or mixing of aerosol types in an atmospheric column as well as the surface albedo or the presence of clouds below an aerosol layer. Furthermore, the instantaneous radiative effect strongly depends on the solar zenith angle.

Keeping all that in mind, we can use, e.g., the study of Kanitz et al. (2013) for some estimates. As shown in this paper, the same aerosol extinction profile leads to changes in the instantaneous TOA solar radiative effect of up to 10 $Wm^{-2}$ depending on the aerosol classification for a specific layer (in this case study for oceanic conditions, a lofted layer with an AOT of 0.2 of either dust, smoke, or a mixture of both above a marine boundary layer with an AOT of 0.2). Table 2 of the paper shows that a variation of asymmetry parameter or SSA of $\pm$ 5 % in one of the layers results in $\pm$ 5–10 $Wm^{-2}$ in TOA solar aerosol radiative effect. Thus, we can conclude that a proper aerosol classification is essential for meeting the radiation closure goals of EarthCARE and that variations of microphysical properties within an aerosol class can be tolerated as long as they do not change the radiative properties by more than a few percent. Kanitz et al. (2013) also showed that the TOA radiative effect is smaller than the surface radiative effect, which could help in closure assessments that are supported/validated by more sensitive surface radiation measurements.

Overall, the questions of the referee can only be answered in detail by performing extended closure studies for a broad variety of scenarios, i.e., by comparing radiative transfer calculations based on

EarthCARE aerosol products with BBR measurements and by supporting such assessments with validation activities from ground.

We have added the additional reference and some discussion regarding the requirements in Sect. 2 (lines 74–78) and in Sect. 6.5 (discussion of Fig. 8, lines 519-521).

Reference:

Kanitz, T., Ansmann, A., Seifert, P., Engelmann, R., Kalisch, J., and Althausen, D.: Radiative effect of aerosols above the northern and southern Atlantic Ocean as determined from shipborne lidar observations, J. Geophys. Res. Atmos., 118, 12,556–12,565, 2013. https://doi.org/110.1002/2013JD019750

2. Lines 44 and 525 mention aerosol components associated with aerosol transport models (e.g. sulfate, organic carbon, black carbon). Line 525 mentions that these aerosol components in the CAMS model were mapped to the HETEAC components. How was this mapping done? It would be helpful to know this mapping in order to use the EarthCARE measurements to help evaluate the ability of aerosol transport models to apportion aerosol extinction and AOD to the model aerosol components.

The mapping between the HETEAC components and the CAMS types is presented in Qu et al.: "Numerical Model Generation of Test Frames for Pre-launch Studies of EarthCARE's Retrieval Algorithms and Data Management System" (Atmos. Meas. Tech. Discuss., https://doi.org/10.5194/amt-2022-300, 2022). It should be noted that this mapping was done in an ad hoc fashion with the limited aim of providing "realistic enough" test data and, though it may serve as a useful starting point, the mapping should likely be revisited for the purposes the reviewer has in mind. We added an additional hint to the reference in the text in Sect. 7.1 (line 540).

3. Line 166. Why were only ground-based observations of the lidar ratio and particle linear depolarization used without any airborne measurements?

So far, we used only measurements from networks and campaigns with strong involvement of TROPOS. In these cases, we have direct access to instruments and raw data as well as complete control over data evaluation and quality assurance, i.e., we are sure to follow always the same standards. We did not aim for completeness or high number of data points, but rather for a good representation of different aerosol types. Nevertheless, we see this work as a starting point and will open the collection for contributions from the community in the future. Please, see the paper by Floutsi et al.: "DeLiAn – a growing collection of depolarization ratio, lidar ratio and Ångström exponent for different aerosol types and mixtures from ground-based lidar observations" (Atmos. Meas. Tech. Discuss., https://doi.org/10.5194/amt-2022-306, 2022) for a more detailed description of this work. – No changes made to the text.

4. Figure 2. It seems that the HETEAC fine strongly absorbing component provides an upper bound on the observed lidar ratio, the HETEAC coarse spherical component provides a lower bound on the observed lidar ratio, and both components provide lower bounds on the observed particle linear depolarization. However, there seems to be several observations of particle linear depolarization ratio that exceed the depolarization ratio of the HETEAC coarse non-spherical component. Could/should this component have been chosen such that its depolarization ratio was higher (~30%) to better bound the observed depolarization ratios?

At first glance, this is a very logical question. However, from the practical point of view, it is not advisable to follow this approach, which has to do with the natural variability of pure dust on the one hand and the limits of the scattering model for non-spherical particles on the other hand. While the corner points for the spherical particles follow naturally when realistic size and refractive-index parameters are applied in the Mie calculations, it is very tricky to obtain a physical meaningful model

representation for the dust particles, as discussed in Sect. 6.3.4 and shown in Fig. 5. Therefore, the dust component has been selected such that it stands for realistic (most appropriate) values of refractive index and particle size, and thus also for meaningful radiative properties. Otherwise, we would end up with a worse representation of dust in the model (always keeping in mind that the spheroidal shape model limits the physical representation anyhow).

In addition, one has to think about the consequences for the representation of pure dust and dust-containing mixtures. If we set the component further to the right (i.e., closer to 30 % depolarization), most of the pure dust observations, naturally occurring between 20 % and 30 % depolarization, would have to be represented in the model as a mixture of dust and non-dust particles, with consequences for the radiative properties. In contrast, with the approach of the probability distribution functions (PDFs) for the ATLID target classification (A-TC product, see Fig. 9), it is relatively easy to account for a wider range of natural dust cases by setting the PDF ellipse appropriately around the values defined for the dust component. In this way, best-estimate dust properties can be assigned to all observations that fall into the dust category with a certain probability.

Nevertheless, we also recommend that users adapt the parameters of the dust component as needed, when it is required for certain applications (e.g., regional studies). This can be done either in the model (by varying the refractive index, particle size, or shape) or in the application (e.g., in A-TC by shifting the PDFs). In any case, and in particular when optimal-estimation techniques are applied in retrievals, it is very important to assure a good overlap between the actual observation space (given by the variability in optical parameters, including measurement errors) and the chosen model space (determined by the variation of the microphysical parameters). – No changes made to the text.

5. Table 2. It would be helpful to know the values of lidar ratio, linear particle depolarization ratio, and Angstrom exponent corresponding to the standard Nd:YAG wavelengths (532, 1064 nm) used by many aerosol lidars including CALIOP.

These values (and also those for the MSI wavelengths) are included in the look-up table and provided in the associated Zenodo publication. Please refer also to our reply to comment 10 regarding the limitations of wavelength-dependent lidar-ratio calculations for non-spherical particles. We added a sentence with the reference in the text of Sect. 6.1 (lines 214–215). The reference is also provided at the end of the paper under *Data availability*.

ratio at 355 nm but would be unable to do so at 532 nm? Is this correct? If so, how will this impact the use of the HETEAC scheme when relating ATLID observations of dust to those from CALIOP?

As shown in Fig. 6 and explained in the related text in Sect. 6.3.4, the major issue with the model is the spectral behavior of the backscatter coefficient, which shows large negative Angström exponents (of the order of −1 to −2) for realistic values of the refractive index (for a given size and shape distribution). This spectral slope of the backscatter coefficient leads to a large difference between the lidar ratios at 355 and 532 nm (of about a factor of 2), which is not supported by observations, even if negative Angström exponents (between 0 and −0.75) are often obtained. The following figure illustrates the situation in the S−$\delta$ diagram.

[Figure]

*Figure 1: Simulated values of lidar ratio versus particle linear depolarization ratio at 355 nm (blue symbols) and 532 nm (green symbols) for the spheroid distribution after Koepke et al. (2015) with an effective radius of 1.94 µm and varying complex refractive index as indicated in the figure. The red circles indicate the HETEAC values.*

Here, we can clearly see that for a given size and shape distribution, the lidar ratio at 532 nm is in general considerably smaller than at 355 nm. Even if it is known that the lidar ratio can differ from region to region due to the mineralogical composition of dust, observations show only slightly lower lidar ratios at 532 nm compared to 355 nm of less than 5 sr within a certain region (see Floutsi et al., 2022), what obviously cannot be reflected with the model (except we would reverse the absorption properties and assume low absorption in the blue and high absorption in the green, what is not realistic either). Regarding depolarization, the model correctly indicates higher values at 532 nm than at 355 nm.

As a consequence, when considering lidar observations at both wavelengths, e.g., to relate ATLID and CALIOP measurements or to develop multi-wavelength typing schemes, we cannot simply use the modelled values. Depending on the purpose, it is necessary to adjust the applied parameters with the help of observational values.

We want to emphasize here again that the issue with the spectral behavior of the backscatter coefficient is not just a problem for HETEAC. It is inherent in any retrieval that uses the spheroidal shape model to calculate lidar parameters from indirect observations (e.g., when inferring lidar ratios from Sun photometer measurements). Respective discussion is provided in Sect. 8.4. – No changes made to the text.

11. Figure 7. How does the HETEAC handle lidar observations of lidar ratio and/or particle depolarization that lie outside the triangles of points formed by the HETEAC components?

In principle, HETEAC does not handle lidar observations directly, but provides the baseline for different algorithms. These algorithms must consider the natural variability of aerosol properties and the measurement errors, but also invalid data. In case of the ATLID target classification (A-TC), as

shown in Fig. 9 and explained in the text, the parameter space is divided in six aerosol types. For each of the types, two-dimensional Gaussian PDFs are defined, which fill the entire parameter space. The PDFs assign a probability to each data point for belonging to a certain component or mixture. Similarly, HETEAC can be used to provide a priori information for optimal-estimation methods, which then deal with the variations of the observational data points within pre-defined error limits. In general, EarthCARE Level 2 algorithms apply various quality control mechanisms and flag the data accordingly. As can be seen in Fig. 10, A-TC contains an "out of parameter space" and a "missing data" class to deal with bad data. Thus, when the observation and its associated error places a data point too far from the triangle, the aerosol type is set to unknown. Respective thresholds and settings are configurable, as for all EarthCARE L2 algorithms. Explanations are provided in the context of Sect. 7.2.1; the statement in line 554 is clarified.

12. Figure 9 and Table 6 provide lidar observable values that correspond to the six tropospheric A-TC aerosol types and how these types are comprised of the HETEAC components. Lines 154-155 state that "The EarthCARE aerosol classification scheme shall preserve the aerosol types of CALIPSO as far as possible to allow long-term global investigations over the lifetime of both missions." This is a commendable goal and will be very important as one would expect that researchers will attempt to combine the CALIOP and ATLID measurements for long term records. The six tropospheric A-TC aerosol types seem to coincide with some of the same aerosol types (at least in name) associated with the CALIOP tropospheric aerosol types. For these apparently common types, how do the lidar observables (at 532 nm) associated with the A-TC types coincide with the corresponding CALIOP values observed (particle depolarization) or assumed (lidar ratio) for these associated aerosol types? How do the underlying aerosol properties (e.g. size, SSA, refractive index) for this common aerosol types compare between A-TC and CALIOP aerosol types?

A harmonization of the long-term CALIPSO-EarthCARE data set for both aerosol and cloud observations (and subsequent radiation studies) is an ultimate goal and a challenging task for the atmospheric science community. We do not think that ad hoc approaches, such as a one-to-one relation of the aerosol types, are sufficient in this context, although common definitions of the targets provide already a good basis. For aerosols, we have to consider the wavelength conversion as well as the consequences of typing from either L1 (CALIOP) or L2 data (ATLID). While HETEAC and A-TC are built on intensive, i.e., concentration-independent, properties and do not consider (much) auxiliary information, the selection criteria in the CALIPSO typing scheme make use of the strength of the aerosol signal, the surface type, and the elevation of the layer (Omar et al., 2009; Kim et al., 2018). Therefore, we believe that a good harmonization of the data set will require some additional efforts, in particular the use of CALIPSO Level 2 data for a refinement of the typing as well as the development of type- and location-dependent conversion schemes.

The newest version of the CALIPSO typing scheme for the troposphere considers 7 subtypes, namely dust, marine, polluted continental/smoke, elevated smoke, clean continental, polluted dust, and dusty marine. Although we can relate the A-TC types to the rather similar CALIPSO subtypes (see Table 1 below), we cannot directly compare A-TC and CALIOP optical data. A-TC uses ATLID L2 particle depolarization ratio and lidar ratio at 355 nm together with configurable input parameters for the PDFs defining the types. The CALIOP algorithm applies an estimated particle depolarization ratio (L1 data product, 532 nm) and assumes an initial lidar ratio (at 532 and 1064 nm), which can be further adjusted within certain limits in the L2 retrievals (see Kim et al., 2018, Table 2). To compare the A-TC and CALIOP approaches, we can apply HETEAC to obtain 355-to-532-nm conversion information (the look-up table provides optical data for a variety of wavelengths and mixtures of the four components, https://doi.org/10.5281/zenodo.7732338). However, as mentioned above, care has to be taken for all mixtures containing non-spherical particles, because of the limitations of the spheroid model. For these cases, 355-to-532 nm conversions of the lidar ratio should be adjusted with the help of observations (e.g., by using DeLiAn, Floutsi et al., 2022). In Table 1 below, we relate the A-TC and CALIOP types and show comparisons of lidar ratios at 355 and 532 nm, including DeLiAn

values. When keeping the discussion above in mind, we find a very good agreement between the approaches/assumptions. This gives us confidence that we have a good starting point for harmonizing the long-term data set and we can build on it in future developments.

Regarding the comparison of the underlying microphysical and radiative properties, it seems that the aerosol microphysical model used in the development of the CALIOP typing scheme (Omar et al., 2009) is somewhat outdated. It was used to estimate the lidar ratios to be applied in the CALIOP retrievals at the beginning of the mission. Lidar ratios as well as aerosol types have been modified in the meantime (Kim et al. 2018) and, therefore, do not fully comply with the original model anymore. From the description of the model (Omar et al., 2009, Table 1), we do not directly get the information that is needed for comparisons with HETEAC (e.g., effective radius, SSA), i.e., further calculations are necessary. Oikawa et al. (2013, 2018) have used the model to estimate the global shortwave radiative forcing and present the SSA values that follow from the microphysical data (see their Table 1). However, they argue that "the SSA derived from the CALIOP aerosol models is not always realistic". Compared to the HETEAC mixtures presented in Table 1 below, the SSA at 532 nm is the same for marine aerosol (0.99) and somewhat higher for dust (0.92 vs 0.90). For all other types, the SSA of the CALIOP model is lower than the one of HETEAC (e.g., 0.93 vs 0.97 for polluted continental and 0.83 vs 0.86 for smoke). Moreover, the effective radius (and thus the asymmetry parameter) following from the bimodal size distributions used in the CALIOP model is for some of the types quite unrealistic and not comparable with HETEAC. In particular, the dust type contains a large fraction of fine-mode particles leading to an effective radius of 0.4 µm (compared to 1.9 µm in HEATEC for the pure coarse mode or 1.2 µm for a mixture with 5 % fine-mode particles). The polluted dust model of CALIOP has nearly the same size distribution and effective radius as the pure dust model, i.e., the effective radius is also smaller than the one of the HETEAC dusty smoke mixture, which is about 0.7 µm. The largest particles of the CALIOP model occur in the clean continental background aerosol with an effective radius of 1.4 µm, while the polluted continental category shows the smallest value of 0.26 µm (comparable to the HETEAC continental pollution with 0.24 µm). From these findings, we conclude that the CALIOP model should be revisited, before it is used for any further studies. We also see how important radiation closure assessments and sub-orbital validation studies will be for the harmonization of the long-term data set, since in addition to the typing we need a good understanding of the aerosol microphysical properties to study aerosol radiative effects.

*Table 1: Lidar ratio values (in sr) applied for ATLID and CALIOP (V4) aerosol typing*

| ATLID Type | A-TC 355 nm (adjustable) | HETEAC* 355 nm | HETEAC* 532 nm | DeLiAn 532 nm | CALIOP Type | Initial Value 532 nm |
|---|---|---|---|---|---|---|
| Marine | 25 | 24 (5:0:90:5) | 23 (5:0:90:5) | 22 (Clean marine) | Clean marine | 23 |
| Dust | 55 | 52 (0:0:5:95) | 30 (0:0:5:95) | 53 / 37 (Sahara / Asia) | Dust | 44 |
| Continental pollution | 55 | 55 (50:5:40:5) | 49 (50:5:40:5) | 47 (Pollution) | Poll. cont. / smoke | 70 |
| Continental pollution | 55 | 55 (50:5:40:5) | 49 (50:5:40:5) | 56 (Europ. bckgr.) | Clean cont. | 53 |
| Smoke | 88 | 84 (30:50:10:10) | 73 (30:50:10:10) | 72 (Smoke) | Elevated smoke | 70 |
| Dusty Smoke | 73 | 72 (5:10:5:80) | 45 (0:10:0:90) | 56 (Dust+smoke) | Polluted dust | 55 |
| Dusty Mix | 43 | 45 (5:5:40:50) | 34 (5:5:40:50) | 32 (Dust+marine) | Dusty marine | 37 |

*HETEAC values for selected volume mixing ratios as indicated
  (weakly abs. fine : strongly abs. fine : spherical coarse : non-spherical coarse)

We added some discussion on the required future efforts in the conclusion and outlook section (lines 757-771, see also reply to comment 3 of Referee #1), but we do not go into the details discussed above, since harmonizing the CALIOP and ATLID data sets is a separate topic, which is beyond the scope of the paper. We will consider all these aspects in future publications.

References (for CALIOP SSA values, not used in the paper):

Oikawa, E., Nakajima, T., Inoue, T., and Winker, D.: A study of the shortwave direct aerosol forcing using ESSP/CALIPSO observation and GCM simulation, Journal of Geophysical Research: Atmospheres, 118, 3687– 3708, 2013.
https://doi.org/10.1002/jgrd.50227

Oikawa, E., Nakajima, T., and Winker, D.: An evaluation of the shortwave direct aerosol radiative forcing using CALIOP and MODIS observations. Journal of Geophysical Research: Atmospheres, 123, 1211–1233, 2018.
https://doi.org/10.1002/2017JD027247

13. Figure 10. This is a nice figure to show an example of the HETEAC aerosol classification and brings to mind a few questions.

a. I don't recall seeing elsewhere in the paper a discussion of the spatial and vertical resolutions of the ATLID retrievals of lidar ratio and linear particle depolarization; what are these resolutions? Presumably the resolutions associated with the lidar ratio retrievals are coarser than the resolutions associated with the retrievals of particle linear depolarization. If this is true, how does the HETEAC classification deal with these different resolutions?

The sophisticated retrieval scheme is described in the respective paper on ATLID profile products by Donovan et al.: "The ATLID L2a profile processor (A-AER, A-EBD, A-TC and A-ICE products)", which will be submitted to the Special Issue soon. The algorithm works on a layer basis with different horizontal resolutions adapted to the specific scene. All products are provided at high (1 km), medium (e.g., 50 km), and low (150 km) horizontal resolution, and these three resolutions are always the same for all products (to make the products usable in combination). We added some text in Sect. 7.2.1 (lines 567-572, see also reply to comment 2 of Referee #1).

b. There are gaps (white areas) in the lidar ratio and depolarization images on the right. Are these gaps because the aerosol loading was too small for trustworthy measurements? If so, what are the minimum aerosol extinction values for which there will be expected retrievals of lidar ratio and linear particle depolarization? Likewise, what are the minimum aerosol extinction values required for the HETEAC aerosol classification?

In case of the depolarization the white gaps show regions for which the cross-polar channel showed a negligible signal with respect to the co-polar signal. White in this case therefore does not indicate no usable signal.

The white gaps in the lidar ratio can originate for two reasons. At first, the ATLID feature mask might have indicated that there are not enough co-polar Mie signals to indicate the existence of aerosol/ice cloud features. This will ensure that no retrieval is attempted. Secondly, the ATLID profile algorithm may not be able to perform a trustworthy retrieval.

Once there is a retrieval of lidar ratio and its error (and therefore the extinction), a target will be assigned. As mentioned above (see reply to comment 2 of Referee #1), it is expected that useful layer-average lidar-ratio retrievals (SNR of the retrieval better than 100 %) can be done for minimum extinction values on the order of 5.0e-6 m$^{-1}$ on the 50-100 km horizontal scale. We included this information in the additional text in Sect. 7.2.1 (lines 567-572, see also reply to comment 2 of Referee #1).

c. Looking again at the images on the right, there are more white areas (no retrievals?) associated with linear particle depolarization than for lidar ratio. Why? (I would have expected the opposite). Does the HETEAC algorithm still attempt to perform a classification if there is a retrieval of only one of two lidar observables?

The white areas correspond both to areas where the retrieved depolarization ratio is zero or below. The model-truth linear depolarization ratio for the aerosol in this scene is close to zero. Details can be found in the paper by Donovan et al.: "The Generation of EarthCARE L1 Test Data sets Using Atmospheric Model Data Sets" (Atmos. Meas. Tech. Discuss., https://doi.org/10.5194/egusphere-2023-384, 2023).

The aerosol classification procedure uses both the lidar ratio and the depolarization ratio along with their respective uncertainties. So, in principle, a retrieval of both observables is always technically available, however, one of the observables may have a very high error associated with it. In this case, the most-probable classification will still be selected. However, if the assesses probability is below a certain threshold, the classification will be set to "unknown". – No changes made to the text.

d. The right side of the top right image shows relatively high values (70-80 sr) of the lidar ratio and low values of aerosol depolarization around 5 km. Figure 2a shows that these lidar ratios and depolarization ratios are associated are smoke and/or mixtures that contain smoke. However, the target classification shown in the bottom right seems to show this area is dominated by continental pollution with little, if any, smoke or smoke mixtures. Why?

After the retrieval of the target classification, a horizontal consistency check is performed using an 'hybrid median' type edge detection, gap filling procedure. This procedure uses the target type occurrence instead of the median as described in the AC-TC and A-PRO papers. This post-filtering will harmonize the types more than would be expected by the lidar ratio figure. More detail on this procedure can be found in Irbah et al.: "The classification of atmospheric hydrometeors and aerosols from the EarthCARE radar and lidar: the A-TC, C-TC and AC-TC products'' (Atmos. Meas. Tech. Discuss., https://doi.org/10.5194/egusphere-2022-1217, 2022). – No changes made to the text.

14. Line 565. Can the column integrated aerosol classification probabilities be illustrated for the example shown in Figure 10?

Such a figure is provided in the paper on the ATLID layer products by Wandinger et al.: "Cloud top heights and aerosol layer properties from EarthCARE lidar observations: the A-CTH and A-ALD products", which will be submitted to the Special Issue soon. We added a remark in the text (line 585-586).

15. Line 678. Suggest changing to "Ground-based and airborne measurements that measure..."

We added the airborne measurements in the sentence, now appearing in Sect. 8.1 (line 659).

**Other changes not related to the referee comments**

We changed the wording regarding the development of the experimental basis at the end of Sect. 4 to better describe our goals (lines 165-170).

We added the reference to the DeLiAn dataset:

[revised manuscript text omitted]